# Unsupervised Federated Graph Matching with Graphlet Feature Extraction and Separate Trust Region

## Abstract

Graph matching in the setting of federated learning is still an open problem. This paper proposes an unsupervised federated graph matching algorithm, UFGM, for inferring matched node pairs on different graphs across clients while maintaining privacy requirement, by leveraging graphlet theory and trust region optimization. First, the nodes' graphlet features are captured to generate pseudo matched node pairs on different graphs across clients as pseudo training data for tackling the dilemma of unsupervised graph matching in federated setting and leveraging the strength of supervised graph matching. An approximate graphlet enumeration method is proposed to sample a small number of graphlets and capture nodes' graphlet features. Theoretical analysis is conducted to demonstrate that the approximate method is able to maintain the quality of graphlet estimation while reducing its expensive cost. Second, we propose a separate trust region algorithm for pseudo supervised federated graph matching while maintaining the privacy constraints. In order to avoid expensive cost of the second-order Hessian computation in the trust region algorithm, we propose two weak quasi-Newton conditions to construct a positive definite scalar matrix as the Hessian approximation with only first-order gradients. We theoretically derive the error introduced by the separate trust region due to the Hessian approximation and conduct the convergence analysis of the approximation method.

## 1 Introduction

Federated graph learning (FGL) is a promising paradigm that enables collaborative training of shared machine learning models over large-scale distributed graph data, while preserving privacy of local data (Zheng et al., 2020; Chen et al., 2021; Zhang et al., 2021a). Only recently, researchers have started to attempt to study the FGL problems (Suzumura et al., 2019; Mei et al., 2019; Zhou et al., 2020b; Jiang et al., 2020; Wang et al., 2020a; Chen et al., 2021; Ke & Honorio, 2021; Wu et al., 2021; Wang et al., 2021a; He et al., 2021b;c). Most of them concentrate on node classification (Zhang et al., 2021b; Wang et al., 2022a; Chen et al., 2022a; Baek et al., 2022; Xie et al., 2023; Zhang et al., 2023; Li et al., 2023), graph classification (Xie et al., 2021; He et al., 2021a; Tan et al., 2022; Wang et al., 2022b), network embedding (Ni et al., 2021; Zhang et al., 2022; Hu et al., 2023; Zhu et al., 2023), and link prediction (Chen et al., 2022c; Baek et al., 2022). Graph matching (i.e., network alignment) is one of the most important research topics in the graph domain, which aims to match the same entities (i.e., nodes) across two or more graphs (Zhang & Yu, 2015; Zhang et al., 2015; Liu et al., 2016; 2017; Malmi et al., 2017; Vijayan & Milenkovic, 2018; Nassar et al., 2018; Zhou et al., 2018b; Chu et al., 2019; Wang et al., 2019b). It has been widely applied to many real-world applications ranging from protein network matching in bioinformatics (Kelley et al., 2003; Singh et al., 2008), user account linking in different social networks (Shu et al., 2016; Mu et al., 2016; Zhong et al., 2018; Li et al., 2018; Zhou et al., 2018a; Feng et al., 2019; Li et al., 2019a), and knowledge translation in multilingual knowledge bases (Xu et al., 2019b; Zhu et al., 2019), to geometric keypoint matching in computer vision (Fey et al., 2020).

While the existing techniques have achieved remarkable performance in the above graph learning domains, there is still a paucity of techniques of effective federated graph matching (FGM), which is much more difficult to study. Directly sharing and inferring matched node pairs on different graphs

across clients and local graphs over multiple clients gives rise to a serious privacy leakage concern and thus limits the applicability of graph matching in the centralized setting, such as user account linking in social networks and financial crime detection on transaction networks (Suzumura et al., 2019; Wang et al., 2019a; Zhang et al., 2021a; NSF; IBM), where the social network data and the bank customer and transfer data contain many sensitive information, advocating the invention of novel FGM techniques. In this work, we aim to answer the following questions: (1) How to train effective FGM models on distributed clients with maintaining high matching performance? (2) How to make FGM models with strong privacy protection for cross-client information exchange?

Research activities on centralized graph matching can be classified into two groups: supervised graph matching (Man et al., 2016; Zhou et al., 2018a; Yasar & Çatalyürek, 2018; Li et al., 2019b;a; Chu et al., 2019; Fey et al., 2020) and unsupervised graph matching (Zhou et al., 2018b; Heimann et al., 2018; Zhong et al., 2018; Li et al., 2018; Huynh et al., 2020b). The former utilizes a set of pre-matched node pairs between pairwise graphs belonging to the same entities as training data to learn an effective graph matching model by minimizing the distances (or maximizing the similarities) between the pre-matched node pairs. The latter fails to employ the strength of training data and thus often leads to sub-optimal solutions. However, supervised graph matching using the pre-matched node pairs as the training data is improper for the FGM scenarios due to privacy risks of direct cross-client information exchange when the graphs to be matched are distributed over different clients.

This motivates us to capture nodes' graphlet features to generate pseudo matched node pairs on different graphs across clients as the pseudo training data for leveraging the strength of supervised graph matching. A graphlet is a small graph of size up to $k$ nodes of a larger graph, such as triangle, wedge, or $k$-clique, which describes the local topology of a larger graph. A node's local topology can be measured by a graphlet feature vector, where each component denotes the frequency of one type of graphlets. Thus, a graphlet feature vector is one of node structure representation (Shervashidze et al., 2009; Kondor et al., 2009; Soufiani & Airoldi, 2012; Jin et al., 2018; Tu et al., 2019). It is highly possible that the nodes in different graphs with the small distances regarding their graphlet features correspond to the same entities. Thus, they can be treated as the pseudo matched node pairs for pseudo supervised FGM.

However, graphlet enumeration one by one on large graphs is impossible due to expensive cost. We propose to leverage Monte Carlo Markov Chain (MCMC) technique for sampling a small number of graphlets. The number of graphlet samples is much smaller than that of all graphlets in the graphs, which dramatically improves the efficiency of graphlet enumeration. Theoretical analysis is conducted to demonstrate that the estimated graphlet count based on the MCMC sampling strategy is close to the actual count of all graphlets, which implies that the graphlet samples and all graphlets share similar distributions.

In order to maintain the privacy requirement of federated learning, we first encrypt local raw graph data on each client with a key shared by all clients (not accessed by the server). The encrypted graph data from all clients are accessed by only the server (not by other clients) for matching the graphs with each other. Note that stochastic gradient descent (SGD) optimization widely used in deep learning fails to work on the clients in the FGM, since each client can access only its own local graph data and thus cannot update local loss based on the pseudo matched node pairs. We propose a separate trust region algorithm for pseudo supervised FGM while maintaining the privacy constraints. Specifically, we separate model optimization from model evaluation in the trust region algorithm: (1) the server aggregates the local model parameter $M_b^s$ on each client $s$ into a global model parameter $M_b$ at global iteration $b$, runs and evaluates $M_b$ on the all pseudo training data $\tilde{D}^{st}$ and the encrypted graph data, and computes the individual loss $\mathcal{L}^s(M_b)$, the gradient $\nabla \mathcal{L}^s(M_b)$, and the Hessian $\nabla^2 \mathcal{L}^s(M_b)$ for each client $s$; (2) client $s$ receives its individual $\mathcal{L}^s(M_b)$, $\nabla \mathcal{L}^s(M_b)$, and $\nabla^2 \mathcal{L}^s(M_b)$ from the server and optimizes $M_{b+1}^s$.

Unfortunately, the second-order Hessian computation $\nabla^2 \mathcal{L}^s(M_b)$ in the separate trust region algorithm is time-consuming over large graphs. We propose to explore quasi-Newton conditions to construct a positive definite scalar matrix $\alpha_b \mathbf{I}$, where $\alpha_b \geq 0$ is a scalar and $\mathbf{I}$ is an identify matrix. Client $s$ uses only first-order gradients $\nabla \mathcal{L}^s(M_b)$ to compute the Hessian approximation, i.e., $z^T \nabla^2 \mathcal{L}^s(M_b) z \approx \alpha_b z^T z$. We theoretically derive the error by the separate trust region due to the Hessian approximation and conduct the convergence analysis of the approximation method.

To our best knowledge, this work is the first to offer an unsupervised federated graph matching solution for inferring matched node pairs on different graphs across clients while maintaining the privacy requirement of federated learning, by leveraging the graphlet theory and trust region optimization. Our UFGM method exhibits three compelling advantages: (1) The combination of the unsupervised FGM and the encryption of local raw graph data is able to provide strong privacy protection for sensitive local data; (2) The graphlet feature extraction can leverage the strength of supervised graph matching with the pseudo training data for improving the matching quality; and (3) The separate trust region for pseudo supervised FGM is helpful to enhance the efficiency while maintaining the privacy constraints.

Empirical evaluation on real datasets shows the superior performance of our UFGM model against several state-of-the-art centralized graph matching, federated domain adaption, and FGL methods.

## 2 BACKGROUND

### 2.1 SUPERVISED GRAPH MATCHING

Given a set of $S$ graphs $G = \{G^1, \cdots, G^S\}$. Each graph is denoted as $G^s = (V^s, E^s)$ $(1 \le s \le S)$, where $V^s = \{v_1^s, v_2^s, \cdots\}$ is the set of nodes and $E^s = \{(v_i^s, v_j^s) : 1 \le i, j \le |V^s|, i \ne j\}$ is the set of edges. Each $G^s$ has a binary adjacency matrix $\mathbf{A}^s$, where each entry $\mathbf{A}_{ij}^s = 1$ if there exists an edge $(v_i^s, v_j^s) \in E^s$; otherwise $\mathbf{A}_{ij}^s = 0$. $\mathbf{A}_{i:}^s$ specifies the $i^{th}$ row vector of $\mathbf{A}^s$ and is used to denote the representation of a node $v_i^s$.

The entire training data consist of a set of training data between pairwise graphs, i.e., $D = \{D^{12}, \cdots, D^{1S}, \cdots, D^{(S-1)S}\}$. Each $D^{st}$ $(1 \le s < t \le S)$ specifies a set of pre-matched node pairs $D^{st} = \{(v_i^s, v_j^t) | v_i^s \leftrightarrow v_j^t, v_i^s \in V^s, v_j^t \in V^t\}$, where $v_i^s \leftrightarrow v_j^t$ represents that two nodes $v_i^s$ and $v_j^t$ are the equivalent ones in two graphs $G^s$ and $G^t$ and are treated as the same entity. The objective of supervised graph matching is to utilize $D^{st}$ as the training data to identify the one-to-one matchings between nodes $v_i^s$ and $v_j^t$ in the test data.

Based on structure, attribute, or embedding features, existing efforts often aim to learn an matching function $M$ to map the node pairs $(v_i^s, v_j^t) \in D^{st}$ with different features across two graphs into common space, i.e, minimize the distances (or maximize the similarities) between source nodes $M(v_i^s)$ and target ones $M(v_j^t)$ (Man et al., 2016; Zhou et al., 2018a; Yasar & Çatalyürek, 2018; Li et al., 2019b;a). The node pairs $(v_i^s, v_j^t) \in D^{st}$ with the smallest distances in the test data are selected as the matching results. This work follows these existing efforts to design the loss function.

$$\mathcal{L} = \sum_{s=1}^{S} \sum_{t=s+1}^{S} \mathbb{E}_{(v_i^s, v_j^t) \in D^{st}} \|M(v_i^s) - M(v_j^t)\|_2^2 \tag{1}$$

Graph convolutional networks (GCNs) have demonstrated their superior learning performance in network embedding tasks (Kipf & Welling, 2017). In this paper, if there are no specific descriptions, we utilize the GCNs to learn the embedding representation with the same dimensions of each node $v_i^s$ in each graph $G^s$, based on its original structure features $\mathbf{A}_{i:}^s$. The embedding representation of $v_i^s$ is denoted by $\mathbf{v}_i^s$. Thus, the objective of supervised graph matching is reformulated as follows.

$$\mathcal{L} = \sum_{s=1}^{S} \sum_{t=s+1}^{S} \mathbb{E}_{(v_i^s, v_j^t) \in D^{st}} \|M(\mathbf{v}_i^s) - M(\mathbf{v}_j^t)\|_2^2 \tag{2}$$

### 2.2 FEDERATED GRAPH MATCHING

In this paper, without loss of generality, we assume that each client contains only one local graph in the federated setting, but it is straightforward to extend to the case of multiple local graphs owned by each client. Given $S$ clients with a set of $S$ graphs $\mathcal{G} = \{G^1, \cdots, G^S\}$ and their local training data $D = \{D^{12}, \cdots, D^{1S}, \cdots, D^{(S-1)S}\}$, and a server, federated graph matching (FGM) aims to learn a global graph matching model $M$ on the server by optimizing the problem below.

$$\min_{M \in \mathbb{R}^d} \mathcal{L}(M) = \sum_{s=1}^{S} \mathcal{L}^s(M) = \sum_{s=1}^{S} \sum_{t=s+1}^{S} \frac{N^{st}}{N} L^{st}(M)$$

$$\text{where } L^{st}(M) = \frac{1}{N^{st}} \sum_{(v_i^s, v_j^t) \in D^{st}} l_{ij}^{st}(M) \tag{3}$$

where $l_{ij}^{st}(M) = \|M(\mathbf{v}_i^s) - M(\mathbf{v}_j^t)\|_2^2$ denotes the loss function of the prediction on the pre-matched node pair $(v_i^s, v_j^t) \in D^{st}$ made with $M$. $\mathcal{L}^s(M)$ and $\mathcal{L}(M)$ are the local loss function on client $s$ and the global one respectively. $N^{st} = |D^{st}|$ denotes the size of local training dataset $D^{st}$. $N$ is the size of total training data $D$, i.e., $N = N^{12} + \cdots + N^{1S} + \cdots + N^{(S-1)S}$. A local graph matching model $M^s$ is optimized based on the local loss $\mathcal{L}^s(M)$. In the FGM, $M$ is iteratively updated with the aggregation of all $M^1, \cdot, M^S$ on $S$ clients in each round, i.e., $M = \sum_{s=1}^{S} \sum_{t=s+1}^{S} \frac{N^{st}}{N} M^s$.

Observed from Eq.(3), when calculating the local loss $\mathcal{L}^s(M)$ on client $s$ for optimizing the local model $M^s$, we need to access the pre-matched node pairs $\{v_i^s, v_j^t\} \in D^{st}$ and the graph $G^t$ on client $t$. This operation obviously violates the privacy requirement of federated learning. Thus, it is difficult to utilize the pre-matched node pairs for supervised FGM.

## 3 MONTE CARLO MARKOV CHAIN FOR GRAPHLET FEATURE EXTRACTION

As discussed in the last section, the supervised graph matching usually achieves better performance than the unsupervised one. In addition, supervised FGM may lead to serious privacy concerns. In this work, we explore to capture nodes' graphlet features to generate pseudo matched node pairs on different graphs across clients as the pseudo training data for leveraging the strength of supervised graph matching while keeping the local graph data safe.

In order to prohibit other clients and server from accessing local raw graphs and embedding representations on any client $s$ for maintaining the privacy requirement of FGM, we first utilize an efficient matrix generation method (Randall, 1993) to produce a random nonsingular matrix $\mathbf{K}$ as a key. Each client employs $\mathbf{K}$ to encrypt its network embedding $\hat{\mathbf{v}}_i^s = \mathbf{v}_i^s \mathbf{K}$ from the original one $\mathbf{v}_i^s$ and uses its inverse $\mathbf{K}^{-1}$ to decrypt from $\hat{\mathbf{v}}_i^s$ to $\mathbf{v}_i^s = \hat{\mathbf{v}}_i^s \mathbf{K}^{-1}$. The encrypted $\hat{\mathbf{v}}_i^s$ from all clients will be uploaded to the server for graph matching. It is important that $\mathbf{K}$ is kept secret between senders and recipients. In our setting, $\mathbf{K}$ is shared by all clients, but not accessed by the server.

The first step of graphlet feature extraction is to enumerate all graphlets in a graph $G = (V, E)$. Concretely, let $G_k$ be the set of all $C$ connected induced $k$-subgraphs (with $k$ nodes) in $G$. Let $\mathcal{G}_1, \mathcal{G}_2, \cdots, \mathcal{G}_R$ be all $R$ types of non-isomorphic $k$-graphlets (with $k$ nodes) for which we would like to count. We denote a $k$-subgraph $g \in G_k$ that is isomorphic to a $k$-graphlet $\mathcal{G}_r$ $(1 \leq r \leq R)$ as $g \sim \mathcal{G}_r$. The number of $k$-graphlets of type $r$ in $G$ is equal to

$$n_{kr}(G) = \sum_{g \in G_k} \mathbb{I}(g \sim \mathcal{G}_r) \tag{4}$$

where $\mathbb{I}(\cdot)$ is an indicator function.

However, graphlet enumeration one by one on large-scale graphs is impossible due to expensive cost. We propose a MCMC sampling technique for which one can calculate the stationary distribution $p$ on the $k$-subgraphs in $G_k$. We only sample a small number of $k$-subgraphs $g_{k1}, \cdots, g_{kO}$ in $G$, where the size $O << C$. Then we use Horvitz-Thompson inverse probability weighting to estimate the graphlet counts as follows.

$$\tilde{n}_{kr}(G) = \frac{1}{O} \sum_{o=1}^{O} \frac{\mathbb{I}(g_{ko} \sim \mathcal{G}_r)}{p(g_{ko})} \tag{5}$$

Next, we describe how to expand from 1-subgraphs to $k$ subgraphs in the graphlet enumeration. For any $(k-1)$-subgraph $g_{k-1}$, we expend it to a $k$-subgraph by adding a node from its neighborhood $\mathcal{N}_v(g_{k-1})$ at random in terms of a certain probability distribution, where $\mathcal{N}_v(g_{k-1})$ is the set of all nodes adjacent to a certain node in $g_{k-1}$ but not including all nodes in $g_{k-1}$.

This expansion operation can explore any subgraph in $G_k$. It iteratively builds a $k$-subgraph $g_k$ from a starting node. First, suppose that a starting node $v_1$ is sampled from the distribution $q$, which can be computed from local information. We assume that $q(v) = \frac{f(\deg(v))}{F}$, where $f(x)$ is a certain function (usually a polynomial) and $F$ is a user-defined normalizing factor. Thus, a 1-subgraph $g_1 = \{v_1\}$ is generated. Second, it samples an edge $(v_1, v_2)$ uniformly in $\mathcal{N}_e(g_1)$, where $\mathcal{N}_e(g_1)$ is the set of all edges that connect a node in $g_1$ and a node outside of $g_1$. Thus, a node $v_2$ is then attached to $g_1$, forming a 2-subgraph $g_2 = g_1 \cup v_2 \cup (v_1, v_2)$. Similarly, at each iteration, it samples an edge $(v_i, v_{j+1})$ $(1 \leq i \leq j)$ from $\mathcal{N}_e(g_j)$ uniformly at random and attach the node $v_{j+1}$ to the

subgraph $g_j$, forming a $j + 1$-subgraph $g_{j+1} = g_j \cup v_{j+1} \cup (v_i, v_{j+1})$. After $k - 1$ iterations, we obtain a $k$-subgraph $g_k$. Once $g_k$ has been sampled we need to classify it into a graphlet type, i.e., $g_k \sim \mathcal{G}_r$. The method repeats the above process $O$ times until $O$ $k$-graphlets $g_{k1}, g_{k1}, \cdots, g_{kO}$ are produced.

We conduct the theoretical analysis to evaluate the permanence of our graphlet enumeration based on the MCMC sampling, in terms of the difference between the estimated and actual graphlet counts.

In the estimation $\tilde{n}_{kr}(G)$ in Eq.(5), a key problem is to calculate $p(g_{ko})$. The probability $p(g_k)$ of getting a $k$-subgraph $g_k$ via subgraph expansion from a $(k - 1)$-subgraph $g_{k-1}$ is given by the sum $p(g_k) = \sum_{g_{k-1}} \mathbb{P}(g_k|g_{k-1})p(g_{k-1})$, where the sum is taken over all connected $(k - 1)$-subgraphs $g_{k-1} \subset g_k$, and $\mathbb{P}(g_k|g_{k-1})$ is the probability of getting from $g_{k-1}$ to $g_k$ in the expansion process.

$$p(g_k) = \sum_{g_{k-1} \subset g_k} p(g_{k-1}) \frac{\deg_{g_{k-1}}\left(V_{g_k} - V_{g_{k-1}}\right)}{|\mathcal{N}_e(g_{k-1})|} = \sum_{g_{k-1} \subset g_k} p(g_{k-1}) \frac{|E_{g_k}| - |E_{g_{k-1}}|}{\sum_{v \in V_{g_{k-1}}} \deg(v) - 2|E_{g_{k-1}}|} \tag{6}$$

where for a subgraph $g_k \subseteq G$, $V_{g_k}$ the set of its nodes and $E_{g_k}$ is the set of its edges. $\deg_{g_{k-1}}(V)$ specifies the number of nodes in $g_{k-1}$ that are connected to a node set $V$. $\deg(v)$ denotes the number of associated edges of a node $v$.

In order to calculate $p(g_k)$, we need to consider all possible orderings of nodes in $g_k$. Assume that the original node ordering of $g_k$ via the subgraph expansion is $x_k = \{v_1, v_2, \cdots, v_k\}$. Let $\mathcal{S}(g_k) = [v_1, v_2, \cdots, v_k]$ be the set of all possible node sequences of $x_k$. Notice that an induced subgraph $h_l(x_k) = \{v_1, v_2, \cdots, v_l, x_k, G\}$ of graph $G$ with the first $l$ nodes $\{v_1, v_2, \cdots, v_l\}$ in $x_k$ must be a connected subgraph for any $l$ $(1 \leq l \leq k)$. Thus, we have

$$\mathcal{S}(g_k) = \{[v_1, \ldots, v_k] | \{v_1, \ldots, v_k\} = V_{g_k}, g_k|\{v_1, \ldots, v_l\} \text{is connected}\} \tag{7}$$

The following theorems give an explicit solution of the probability $p(g_k)$ of getting a $k$-subgraph $g_k$ via subgraph expansion and the variance of the estimation $\tilde{n}_{kr}(G)$ of graphlet counts.

**Theorem 1.** *Let $x_k = \{v_1, v_2, \cdots, v_k\}$ be the original node ordering of $g_k$ via the subgraph expansion, $\mathcal{S}(g_k) = [v_1, v_2, \cdots, v_k]$ be the set of all possible node sequences of $x_k$, $x_k[i]$ be the $i^{th}$ node in $x_k$, $F$ be a user-defined normalizing factor in the subgraph expansion, and $h_l(x_k) = \{v_1, v_2, \cdots, v_l, x_k, G\}$ be an induced subgraph of graph $G$ with the first $l$ nodes $\{v_1, v_2, \cdots, v_l\}$ in $x_k$, then the probability of getting a $k$-subgraph $g_k$ via the subgraph expansion is*

$$p(g_k) = \sum_{x_k \in \mathcal{S}(g_k)} \frac{f(\deg(x_k[1]))}{F} \prod_{l=1}^{k-1} \frac{\left|E_{h_{l+1}(x_k)}\right| - \left|E_{h_l(x_k)}\right|}{\sum_{i=1}^{l} \deg(x_k[i]) - 2\left|E_{h_l(x_k)}\right|} \tag{8}$$

**Theorem 2.** *Let $\tilde{n}_{kr}(G) = \frac{1}{O} \sum_{o=1}^{O} \frac{\mathbb{I}(g_{ko} \sim \mathcal{G}_r)}{p(g_{ko})}$ be the estimation of graphlet counts, $d_1, \cdots, d_k$ be the $k$ highest degrees of nodes in $G$, and denote $D = \prod_{l=2}^{k-1}(d_1 + \cdots + d_k)$. If $q$ for sampling the starting node is the stationary distribution of the node random walk, then the upper bound of the variance $\mathrm{Var}(\tilde{n}_{kr}(G))$ is*

$$\mathrm{Var}(\tilde{n}_{kr}(G)) \leq \frac{1}{O} n_{kr}(G) \frac{2|E_G|}{|\mathcal{S}(\mathcal{G}_r)|} D \tag{9}$$

*Please refer to Appendix A.2 for detailed proof of Theorems 1 and 2.*

It is observed that the variance $\mathrm{Var}(\tilde{n}_{kr}(G))$ is small when the distribution of $p(g_k)$ is close to uniform distribution. A larger $p(g_k)$ results in a smaller variance of the estimator. Thus, the variation can be reduced by an appropriate choice of $q$ for sampling the starting node, say a smaller normalizing factor $F$. In this case, the estimated graphlet count $\tilde{n}_{kr}(G)$ is close to the actual count $n_{kr}(G)$, which implies that the graphlet samples and all graphlets share similar distributions.

We capture the graphlet features of a node by computing the frequency of each type of graphlet with size up to $k$ that is associated with this node. For the node pairs between pairwise graphs, we compute the cosine similarity scores based on the graphlet features on all $R$ types of graphlet. The top-$K$ node pairs with the largest similarities between pairwise graphs $G^s$ and $G^t$ are treated as the pseudo matched node pairs and added to the pseudo training data $\tilde{D}^{st}$.

# 4 SEPARATE TRUST REGION FOR UNSUPERVISED FEDERATED GRAPH MATCHING

In this work, according to the graphlet-based pseudo training data $\tilde{D}^{st}$ and the encrypted network embedding $\hat{\mathbf{v}}_i^s$, we propose a separate trust region algorithm for pseudo supervised FGM while maintaining the privacy constraints. Specifically, we separate model optimization from model evaluation in the trust region algorithm: (1) the server aggregates the local model parameter $M_b^s$ on each client $s$ into a global model parameter $M_b$ at global iteration $b$, runs and evaluates $M_b$ on all the pseudo training data $\tilde{D}^{st}$ and the encrypted network embeddings $\hat{\mathbf{v}}_i^s$, and computes the individual loss $\mathcal{L}^s(M_b)$, the gradient $\nabla\mathcal{L}^s(M_b)$, and the Hessian $\nabla^2\mathcal{L}^s(M_b)$ for each client $s$; (2) client $s$ receives its individual $\mathcal{L}^s(M_b)$, $\nabla\mathcal{L}^s(M_b)$, and $\nabla^2\mathcal{L}^s(M_b)$ from the server and optimizes $M_{b+1}^s$.

$$\textbf{Server}: \text{Compute } M_b = \sum_{s=1}^{S}\sum_{t=s+1}^{S}\frac{N^{st}}{N}M_b^s, \ L^{st}(M_b) = \frac{1}{N^{st}}\sum_{(v_i^s,v_j^t)\in\tilde{D}^{st}}\|M_b(\hat{\mathbf{v}}_i^s) - M_b(\hat{\mathbf{v}}_j^t)\|_2^2,$$

(10)

$$\mathcal{L}^s(M_b) = \sum_{t=s+1}^{S}\frac{N^{st}}{N}L^{st}(M_b), \ \nabla\mathcal{L}^s(M_b), \text{ and } \nabla^2\mathcal{L}^s(M_b)$$

$$\textbf{Client s}: \text{Optimize } z^* = \arg\min u_b(z) = \mathcal{L}^s(M_b) + (\nabla\mathcal{L}^s(M_b))^T z + \frac{1}{2}z^T\nabla^2\mathcal{L}^s(M_b)z, \text{ s.t.}\|z\| \le \Delta^s$$

$$\text{Update } M_{b+1}^s = M_b^s + z^*$$

(11)

where $\Delta^s > 0$ is the trust-region radius. $z^*$ is the trust-region step. The individual loss $\mathcal{L}^s(M_b)$ aims to minimize the sum of distance between nodes on client $s$ and nodes on other clients in the pseudo training data $\tilde{D}^{st}$. The node pairs with the smallest distance between pairwise encrypted network embeddings are selected as the matching results.

A key challenge in the separate trust region algorithm is to compute the second-order Hessian computation $\nabla^2\mathcal{L}^s(M_b)$. It is time-consuming over large-scale graph data. We propose to explore quasi-Newton conditions to construct a positive definite scalar matrix $\alpha_b\mathbf{I}$, where $\alpha_b \ge 0$ is a scalar and $\mathbf{I}$ is an identify matrix, as the Hessian approximation with only first-order gradients, i.e., $z^T\nabla^2\mathcal{L}^s(M_b)z \approx \alpha_b z^T z$.

Concretely, the quasi-Newton condition is given as follows.

$$\nabla^2\mathcal{L}^s(M_b)z_b = y_b$$

(12)

where $z_b = M_{b+1} - M_b$ and $y_b = \nabla\mathcal{L}^s(M_{b+1}) - \nabla\mathcal{L}^s(M_b)$. The condition is derived from the following quadratic model.

$$u_{b+1}(z) = \mathcal{L}^s(M_{b+1}) + (\nabla\mathcal{L}^s(M_{b+1}))^T z + \frac{1}{2}z^T\nabla^2\mathcal{L}^s(M_{b+1})z$$

(13)

The quadratic model is an approximation of the objective function at iteration $b+1$ and satisfies the following three interpolation conditions:

$$(1) \ u_{b+1}(0) = \mathcal{L}^s(M_{b+1}), \ \ (2) \ \nabla u_{b+1}(0) = \nabla\mathcal{L}^s(M_{b+1}), \ \ (3) \ \nabla u_{b+1}(-z_b) = \nabla\mathcal{L}^s(M_b) \quad (14)$$

It is difficult to satisfy the quasi-Newton equation in Eq.(12) with a nonsingular scalar matrix (Farid et al., 2010). A recent study introduced a weak condition form by projecting the quasi-Newton equation in Eq.(12) in the direction $z_b$ (J. E. Dennis & Wolkowicz, 1993).

$$z_b^T\nabla^2\mathcal{L}^s(M_{b+1})z_b = z_b^T y_b$$

(15)

The choice of $z_b$ may influence the quality of the curvature information provided by the weak quasi-Newton condition. Another weak condition is directly derived from an interpolation emphasizing more on function values rather than from the projection of the quasi-Newton condition (xiang Yuan, 1991).

$$u_{b+1}(-z_b) = \mathcal{L}^s(M_b)$$

(16)

By combining sub-conditions (1) and (2) in Eq.(14) and replacing (3) with Eq.(16), we can get another weak quasi-Newton condition.

$$z_b^T\nabla^2\mathcal{L}^s(M_{b+1})z_b = 2\left(\mathcal{L}^s(M_b) - \mathcal{L}^s(M_{b+1}) + z_b^T\nabla\mathcal{L}^s(M_{b+1})\right)$$

(17)

By integrating two types of weak quasi-Newton conditions together, we have a generalized weak quasi-Newton condition.

$$
\begin{aligned}
z_b^T \nabla^2 \mathcal{L}^s(M_{b+1}) z_b &= (1-\omega) z_b^T y_b + \omega \left[ 2 \left( \mathcal{L}^s(M_b) - \mathcal{L}^s(M_{b+1}) \right) + 2 z_b^T \nabla \mathcal{L}^s(M_{b+1}) \right] \\
&= z_b^T y_b + \omega \left[ 2 \left( \mathcal{L}^s(M_b) - \mathcal{L}^s(M_{b+1}) \right) + \left( \nabla \mathcal{L}^s(M_b) + \nabla \mathcal{L}^s(M_{b+1}) \right)^T z_b \right]
\end{aligned}
\tag{18}
$$

where $\omega \geq 0$ is the weight. If $\nabla^2 \mathcal{L}^s(M_{b+1})$ is set to be a scalar matrix $\alpha_{b+1}^*(\omega)\mathbf{I}$, then we have

$$
\alpha_{b+1}(\omega) = \frac{z_b^T y_b + \omega \left[ 2 \left( \mathcal{L}^s(M_b) - \mathcal{L}^s(M_{b+1}) \right) + \left( \nabla \mathcal{L}^s(M_b) + \nabla \mathcal{L}^s(M_{b+1}) \right)^T z_b \right]}{z_b^T z_b}
\tag{19}
$$

The following theorems derive the error introduced by the separate trust region due to the Hessian approximation and conduct the convergence analysis of the approximation method.

**Theorem 3.** *Let $d$ be the dimension of the flattened $M_{b+1}$, $\otimes$ be an appropriate tensor product, $\mathcal{A}_{b+1} \in \mathbb{R}^{d \times d \times d}$ and $\mathcal{B}_{b+1} \in \mathbb{R}^{d \times d \times d \times d}$ are the tensors of $\mathcal{L}^s(M_{b+1})$ at iteration $b+1$ satisfying*

$$
\mathcal{A}_{b+1} \otimes z_b^3 = \sum_{i,j,k=1}^{d} \frac{\partial^3 \mathcal{L}^s(M_{b+1})}{\partial M^i \partial M^j \partial M^k} z_b^i z_b^j z_b^k
\tag{20}
$$

*and*

$$
\mathcal{B}_{b+1} \otimes z_b^4 = \sum_{i,j,k,l=1}^{d} \frac{\partial^4 \mathcal{L}^s(M_{b+1})}{\partial M^i \partial M^j \partial M^k \partial M^l} z_b^i z_b^j z_b^k z_b^l.
\tag{21}
$$

*Suppose that $\mathcal{L}^s(M_{b+1})$ is sufficiently smooth, if $\|z_b\|$ is small enough, then we have*

$$
z_b^T \nabla^2 \mathcal{L}^s(M_{b+1}) z_b - \alpha_{b+1}(\omega) z_b^T z_b = \left( \frac{1}{2} - \frac{\omega}{6} \right) \mathcal{A}_{b+1} \otimes z_b^3 - \left( \frac{1}{6} - \frac{\omega}{12} \right) \mathcal{B}_{b+1} \otimes z_b^4 + \mathcal{O}\left( \|z_b\|^5 \right)
\tag{22}
$$

**Theorem 4.** *Suppose $\|\nabla \mathcal{L}^s(M_b)\| \neq 0$, the solution $z_b$ of the separate trust region optimization $\arg\min u_b(z) = \mathcal{L}^s(M_b) + (\nabla \mathcal{L}^s(M_b))^T z + \frac{1}{2} z^T \nabla^2 \mathcal{L}^s(M_b) z$, s.t.$\|z\| \leq \Delta^s$ in Eq.(11) satisfies*

$$
u_b(0) - u_b(z_b) \geq \frac{1}{2} \|\nabla \mathcal{L}^s(M_b)\| \min \left\{ \Delta^s, \frac{\|\nabla \mathcal{L}^s(M_b)\|}{\alpha_b} \right\}
\tag{23}
$$

*Please refer to Appendix A.2 for detailed proof of Theorems 3 and 4.*

Finally, the separate trust region based on two weak quasi-Newton conditions is given below.

$$
z^* = \arg\min u_b(z) \approx \mathcal{L}^s(M_b) + (\nabla \mathcal{L}^s(M_b))^T z + \frac{1}{2} \alpha_b(\omega) z^T z, \text{ s.t.} \|z\| \leq \Delta^s
\tag{24}
$$

## 5 EXPERIMENTAL EVALUATION

In this section, we have evaluated the performance of our UFGM model and other comparison methods for federated graph matching over serval representative federated graph datasets to date. We show that UFGM with graphlet feature extraction and separate trust region is able to achieve higher matching accuracy and faster convergence in federated settings against several state-of-the-art centralized graph matching, ~~federated graph learning~~ and federated domain adaption methods.

**Datasets.** We focus on three representative graph learning benchmark datasets: social networks (SNS) (Zhang et al., 2015), protein-protein interaction networks (PPI) (Zitnik & Leskovec, 2017), and DBLP coauthor graphs (DBLP) (DBL). Without loss of generality, we assume that each client contains only one local graph in the federated setting. For the supervised learning methods, the training data ratio over the above three datasets is all fixed to 20%. We train the models on the training set and test them on the test set for three datasets. The detailed descriptions of the federated datasets are presented in Appendix A.5.

**Baselines.** To our best knowledge, this work is the first to offer an unsupervised federated graph matching solution for inferring matched node pairs on different graphs across clients while maintaining the privacy requirement of federated learning, by leveraging the graphlet theory and trust region optimization. Thus, we choose three types of baselines that are most close to the task of federated graph matching: centralized graph matching, ~~federated graph learning~~ and federated domain adaption. We compare the UFGM model with six state-of-the-art centralized graph matching models: **NextAlign** (Zhang et al., 2021c), **NetTrans** (Zhang et al., 2020), **CPUGA** (Pei et al.,

Table 1: Final performance on SNS

| Type | Algorithm | $Hits@1$ | $Hits@5$ | $Hits@10$ | $Hits@50$ | $Loss$ |
|---|---|---|---|---|---|---|
| Centralized Graph Matching | NextAlign | **0.430** | **0.512** | **0.571** | **0.635** | 2.149 |
| | NetTrans | 0.379 | 0.439 | 0.447 | 0.496 | 1.611 |
| | CPUGA | 0.230 | 0.238 | 0.252 | 0.297 | 2.551 |
| | ASAR-GM | 0.199 | 0.229 | 0.252 | 0.337 | 1.410 |
| | SIGMA | 0.220 | 0.232 | 0.253 | 0.262 | 1.330 |
| | SeedGNN | 0.319 | 0.340 | 0.342 | 0.388 | 2.919 |
| Federated Domain Adaption | DualAdapt | 0.001 | 0.002 | 0.002 | 0.002 | 2.049 |
| | EFDA | 0.001 | 0.001 | 0.002 | 0.002 | 3.427 |
| | WSDA | 0.003 | 0.005 | 0.007 | 0.011 | 5.129 |
| | FedKA | 0.001 | 0.001 | 0.010 | 0.013 | 3.715 |
| | UFGM | 0.371 | 0.440 | 0.411 | 0.459 | **0.501** |

Table 2: Final performance on PPI

| Type | Algorithm | $Hits@1$ | $Hits@5$ | $Hits@10$ | $Hits@50$ | $Loss$ |
|---|---|---|---|---|---|---|
| Centralized Graph Matching | NextAlign | **0.951** | **0.962** | **0.972** | **0.979** | 2.115 |
| | NetTrans | 0.921 | 0.932 | 0.958 | 0.960 | 1.571 |
| | CPUGA | 0.248 | 0.392 | 0.433 | 0.563 | 2.598 |
| | ASAR-GM | 0.299 | 0.394 | 0.453 | 0.668 | 1.699 |
| | SIGMA | 0.499 | 0.560 | 0.633 | 0.782 | 1.652 |
| | SeedGNN | 0.884 | 0.943 | 0.959 | 0.960 | 3.039 |
| Federated Domain Adaption | DualAdapt | 0.006 | 0.006 | 0.007 | 0.011 | 2.106 |
| | EFDA | 0.007 | 0.011 | 0.014 | 0.029 | 3.249 |
| | WSDA | 0.009 | 0.011 | 0.013 | 0.016 | 2.746 |
| | FKA | 0.005 | 0.006 | 0.006 | 0.008 | 2.227 |
| | UFGM | 0.771 | 0.880 | 0.902 | 0.930 | **0.659** |

2022), **ASAR-GM** (Ren et al., 2022), **SeedGNN** (Yu et al., 2022), and **SIGMA** (Li et al., 2022), ~~six representative federated graph learning architectures: **FedGraphNN** (He et al., 2021a), **FKGE** (Peng et al., 2021), **SpreadGNN** (He et al., 2022), **SFL** (Chen et al., 2022b), **FederatedScope-GNN** (Wang et al., 2022b), and **FedStar** (Tan et al., 2022),~~ and four recent federated domain adaption methods: **DualAdapt** (Peng et al., 2020), **EFDA** (Kang et al., 2022), **WSDA** (Jiang & Koyejo, 2023), and **FedKA** (Sun et al., 2022). The detailed descriptions of the baselines are presented in Appendix A.5.

**Evaluation metrics.** By following the same settings in two representative graph matching models (Yasar & Çatalyürek, 2018; Fey et al., 2020), We employ a popular measure, $Hits@K$, to evaluate and compare our UFGM model to previous lines of work, where $Hits@K$ measures the proportion of correctly matched nodes ranked in the top-$K$ list. A larger $Hits@K$ value indicates a better graph matching result. We use final $Hits@K$ to evaluate the quality of the federated federated learning algorithms. In addition, we plot the measure curves regarding $Hits@K$ and Loss Function Values ($Loss$) with increasing rounds to verify the convergence of different federated learning methods: (Karimireddy et al., 2020; Mitra et al., 2021; Liu et al., 2020; Reddi et al., 2021; Karimireddy et al., 2021; Wang et al., 2021b). A smaller Loss score shows a better federated learning result.

**Final $Hits@K$ and $Loss$ on SNS and PPI.** Tables 1 and 2 show the quality of six centralized graph matching~~, six federated graph learning,~~ and four federated domain adaption algorithms over SNS and PPI respectively. We have observed that our UFGM federated graph matching solution outperforms all the competitors of ~~federated graph learning and~~ federated domain adaption in most experiments. UFGM achieves the highest $Hits@K$ values ($> 0.771$ over SNS and $> 0.371$ on PPI respectively) and the lowest $Loss$ values ($= 0.659$ over SNS and $= 0.501$ on PPI respectively), which are better than other four baseline methods in all tests. In addition, the $Hits@K$ scores achieved by UFGM is close or much better than the centralized graph matching method. Compared with the best centralized graph matching method, NextAlign, the $Hits@1$, $Hits@5$, $Hits@10$, and

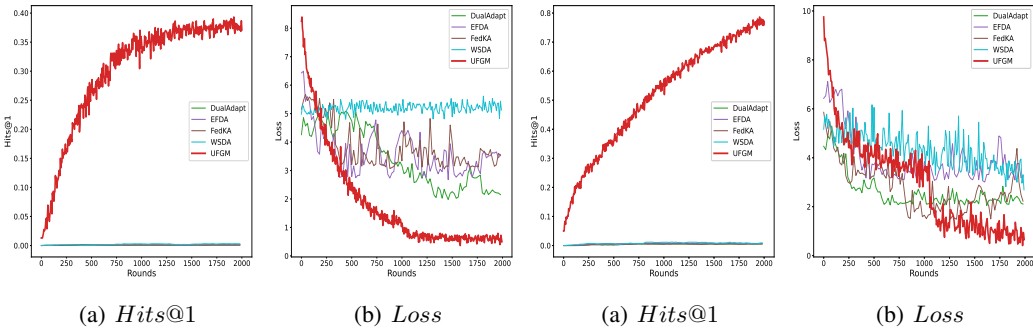

| (a) $Hits@1$ | (b) $Loss$ | (a) $Hits@1$ | (b) $Loss$ |

Figure 1: Convergence on SNS    Figure 2: Convergence on PPI

$Hits@50$ scores by UFGM are only 15.3% lower respectively. A reasonable explanation is that the combination of graphlet feature extraction, separate trust region, and pseudo supervised learning is able to achieve higher matching accuracy and faster convergence in federated settings. In addition, the promising performance of UFGM over both datasets implies that UFGM has great potential as a general federated graph matching solution over federated datasets, which is desirable in practice.

$Hits@K$ **Convergence on SNS and PPI.** Figures 1 and 2 exhibit the $Hits@K$ curves of five federated learning models for graph matching over SNS and PPI respectively. It is obvious that the performance curves by federated learning algorithms initially keep increasing with training rounds and remains relatively stable when the curves are beyond convergence points, i.e., turning points from a sharp $Hits@K$ increase to a flat curve. This phenomenon indicates that most federated learning algorithms are able to converge to the invariant solutions after enough training rounds. However, among ~~six federated graph learning and~~ four federated domain adaption approaches, our UFGM method can significantly speedup the convergence on two datasets in most experiments, showing the superior performance of UFGM in federated settings. Compared to the learning results by other federated learning models, based on training rounds at convergence points, UFGM, on average, achieves 31.8% and 35.4% convergence improvement on two datasets respectively.

$Loss$ **Convergence on SNS and PPI.** Figures 1 and 2 also present the $Loss$ curves achieved by five federated learning models on two datasets respectively. We have observed that the reverse trends, in comparison with the $Hits@K$ curves. In most experiments, our UFGM is able to achieve the fastest convergence, especially, UFGM can converge around 1,000 training rounds and then always keep stable on two datasets. A reasonable explanation is that UFGM fully utilizes the proposed graphlet feature extraction techniques to generate the pseudo training data and employ the strength of supervised graph matching for accelerating the training convergence.

## 6 CONCLUSIONS

In this work, we have proposed an unsupervised federated graph matching algorithm. First, an approximate graphlet enumeration method is proposed to capture nodes' graphlet features to generate pseudo matched node pairs as pseudo training data. Second, a separate trust region algorithm is proposed for pseudo supervised federated graph matching while maintaining the privacy constraints. Finally, empirical evaluation on real datasets demonstrates the superior performance of our UFGM.

## 7 REPRODUCIBILITY STATEMENT

We include the citations and URLs of all datasets used in this work and all codes of third-party baselines in Sections 5 and A.5. Since the datasets used are all public datasets and our methodologies, the experiment environment, the datasets, the training strategies, the baselines, the implementation details, and the hyperparameter settings are explicitly described in Section 3, 4, 5, and A.5, our codes and experiments can be easily reproduced on top of a GPU server. We promise to release our open-source codes on GitHub and maintain a project website with detailed documentation for long-term access by other researchers and end-users after the paper is accepted.

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

# A APPENDIX

## A.1 RELATED WORK

**Centralized Graph Matching.** Graph matching, also well known as network alignment, which aims to identify the same entities (i.e., nodes) across multiple graphs, has been a heated topic in recent years (Chu et al., 2019; Xu et al., 2019a; Wang et al., 2020f; Chen et al., 2020a;c; Zhang & Tong, 2016; Mu et al., 2016; Heimann et al., 2018; Li et al., 2019b; Fey et al., 2020; Qin et al., 2020; Feng et al., 2019; Ren et al., 2020). Research activities can be classified into three broad categories. (1) Topological structure-based techniques, which rely on only the structural information of nodes to match two or multiple graphs, including CrossMNA (Chu et al., 2019), MOANA (Zhang et al., 2019), GWL (Xu et al., 2019a), DPMC (Wang et al., 2020f), MGCN (Chen et al., 2020a), GraphSim (Bai et al., 2020), ZAC (Wang et al., 2020c), GRAMPA (Fan et al., 2020), CONE-Align (Chen et al., 2020c), DeepMatching (Wang et al., 2020b), Exact Graph Matching (Rácz & Sridhar, 2021), qc-DGM (Gao et al., 2021), OTTER (Weighill et al., 2021), IA-GM (Zhao et al., 2021), GMTracker (He et al., 2021d), Proxy Graph Matching (Tan et al., 2021), Fusion Moves for Graph Matching (Hutschenreiter et al., 2021), D-GAP (Lyu et al., 2022), CPUGA (Pei et al., 2022), CAPER (Zhu et al., 2022a); (2) Structure and/or attribute-based approaches, which utilize highly discriminative structure and attribute features for ensuring the matching effectiveness, such as FINAL (Zhang & Tong, 2016), ULink (Mu et al., 2016), gsaNA (Yasar & Çatalyürek, 2018), RE-GAL (Heimann et al., 2018), SNNA (Li et al., 2019b), CENALP (Du et al., 2019), GAlign (Huynh et al., 2020b), Deep Graph Matching Consensus (Fey et al., 2020), CIE (Yu et al., 2020), RE (Zhou et al., 2020c), Meta-NA (Zhou et al., 2020a), G-CREWE (Qin et al., 2020), GA-MGM (Wang et al., 2020d), EAGM (Qu et al., 2021), DLGM (Yu et al., 2021), SIGMA (Liu et al., 2021), CGMN (Jin et al., 2022), FOTA (Liu et al., 2022b), SCGM (Liu et al., 2022a), and Grad-Align+ (Park et al., 2022); (3) Heterogeneous methods employ heterogeneous structural, content, spatial, and temporal features to further improve the matching performance, including SCAN-PS (Zhang et al., 2013), MNA (Kong et al., 2013), HYDRA (Liu et al., 2014), COSNET (Zhang et al., 2015), Factoid Embedding (Xie et al., 2018), HEP (Zheng et al., 2018), LHNE (Wang et al., 2019c), ActiveIter (Ren et al., 2019), NAME (Zhou et al., 2019), TransLink (Zhou & Fan, 2019), DPLink (Feng et al., 2019), DETA (Meng et al., 2019). BANANA (Ren et al., 2020), SAUIL (Qiao et al., 2020). GCAN (Jiang et al., 2022), and Deep Multi-Graph Matching (Ye et al., 2022); Several papers review key achievements of graph matching across online information networks including state-of-the-art algorithms, evaluation metrics, representative datasets, and empirical analysis (Shu et al., 2016; Guzzi & Milenkovic, 2018; Huynh et al., 2020a; Vijayan et al., 2020; Yan et al., 2020; Zhang & Tong, 2020; Haller et al., 2022). It has been widely applied to many real-world applications, including protein network alignment in bioinformatics (Liu et al., 2017; Vijayan et al., 2020), user account linking in multiple social networksShu et al. (2016); Mu et al. (2016); Feng et al. (2019), object matching in computer vision (Fey et al., 2020; Wang et al., 2020c;e; Yang et al., 2020), knowledge translation in multilingual knowledge bases (Xu et al., 2019b; Zhu et al., 2019; Sun et al., 2020; Wu et al., 2020; Zhu et al., 2022b; Chakrabarti et al., 2022; Liu et al., 2022c; Guo et al., 2022; Zhu et al., 2022b; Xin et al., 2022) and text matching (Chen et al., 2020b).

**Federated Graph Learning.** With the increasing privacy awareness, commercial competition, and regulation restrictions, real-world graph data is often generated locally and remains distributed graphs of multiple data silos among a large number of clients (Zheng et al., 2020; Chen et al., 2021; Zhang et al., 2021a). Federated graph learning (FGL) is a promising paradigm that enables collaborative training of shared machine learning models over large-scale distributed graph data, while preserving privacy of local data. Based on how graph data can be distributed across clients, existing FGL techniques on machine unlearning can be broadly classified into three categories below. (1) Graph-level FGL: each client possesses a set of graphs and all clients collaborate to train a shared model to predict graph properties, including (Xie et al., 2021; He et al., 2021a; Zhang et al., 2022; Chen et al., 2022c; Tan et al., 2022; Hu et al., 2023; Qu et al., 2023). Typical graph-level FGL task is graph classification/regression, which have been applied multiple domains, such as molecular property prediction (Xie et al., 2021; He et al., 2022) and brain network analysis (Bayram & Rekik, 2021); (2) Subgraph-level FL: each client contains a subgraph of a global graph, a part of node features, and a part of FGL model (Zhang et al., 2021b; Ni et al., 2021; Wang et al., 2022a; Chen et al., 2022a; Baek et al., 2022; Xie et al., 2023; Zhang et al., 2023; Li et al., 2023; Zhu et al., 2023; Tian et al., 2023). The clients aim to collaboratively train a global model with the partial features

and subgraphs to predict node properties. Typical graph-level FGL task is node classification and link prediction; (3) Node-level FGL: the clients are connected by a graph and thus each of them is treated as a node (Lalitha et al., 2019; Meng et al., 2021; Caldarola et al., 2021; Rizk & Sayed, 2021). Namely, the clients, rather than the data, are graph-structured. For example, each client performs learning with its own data and they exchange data through the communication graph (Lalitha et al., 2019; Meng et al., 2021). The server maintains the graph structure and uses a GNN to aggregate information (either models or data) collected from the clients (Caldarola et al., 2021; Rizk & Sayed, 2021).

A recent work studied the problem of federated knowledge graphs embedding with a byproduct of knowledge graph alignment (Peng et al., 2021). It exploits adversarial generation between pairs of knowledge graphs to translate identical entities and relations of different domains into near embedding spaces. To our best knowledge, this workThis work is the first to has the potential to tackle the problem of general federated graph matching. However, it is a supervised learning method with aligned entities and relations as training data. In addition, it is possible that neural models may memorize inputs and reconstruct inputs from corresponding outputs (Carlini et al., 2021). The method exchanges the embeddings of entities and relations between clients and server. Adversarial samples and gradients are interchanged among the clients. Although a host client cannot access the embeddings of the other's, the exchange of translational mapping matrices (1.e., the gradients in the generators of the other clients) makes it possible for the host client to reconstruct the former's embeddings with the inverse of translational mapping matrices. The combination of the above two properties dramatically limits the applicability of the method in real scenarios. This work is the first to offer an unsupervised federated graph matching solution for inferring matched node pairs on different graphs across clients while maintaining the privacy requirement of federated learning, by leveraging the graphlet theory and trust region optimization.

## A.2 Proof of Theorems

**Theorem 1.** *Let $x_k = \{v_1, v_2, \cdots, v_k\}$ be the original node ordering of $g_k$ via the subgraph expansion, $\mathcal{S}(g_k) = [v_1, v_2, \cdots, v_k]$ be the set of all possible node sequences of $x_k$, $x_k[i]$ be the $i^{th}$ node in $x_k$, $F$ be a user-defined normalizing factor in the subgraph expansion, and $h_l(x_k) = \{v_1, v_2, \cdots, v_l, x_k, G\}$ be an induced subgraph of graph $G$ with the first $l$ nodes $\{v_1, v_2, \cdots, v_l\}$ in $x_k$, then the probability of getting a $k$-subgraph $g_k$ via the subgraph expansion is*

$$p(g_k) = \sum_{x_k \in \mathcal{S}(g_k)} \frac{f(\deg(x_k[1]))}{F} \prod_{l=1}^{k-1} \frac{\left|E_{h_{l+1}(x_k)}\right| - \left|E_{h_l(x_k)}\right|}{\sum_{i=1}^{l} \deg(x_k[i]) - 2\left|E_{h_l(x_k)}\right|} \tag{25}$$

*Proof.* *We can consider a subgraph expansion process as a way of sampling a sequence $x_k = \{v_1, v_2, \cdots, v_k\}$, ordered from the first node sampled to the last one, that is then used to generate a $k$-subgraph $g_k$. Denote the set of such sequences as $V_G^k$. Let $h_l = \{v_1, v_2, \cdots, v_l\}$ is a $l$-subgraph of graph $G$ obtained by the subgraph expansion process on step $l$. The probability of sampling node $v_{l+1}$ on the step $l+1$ to produce a $(l+1)$-subgraph $h_{l+1} = \{v_1, v_2, \cdots, v_l, v_{l+1}\}$ is equal to*

$$\mathbb{P}(v_{l+1} \mid h_l) = \frac{\deg_{h_l}(v_{l+1})}{|\mathcal{N}_e(h_l)|} = \frac{\left|E_{h_{l+1}}\right| - |E_{h_l}|}{\sum_{i=1}^{l} \deg(v_i) - 2|E_{h_l}|} \tag{26}$$

*where $\mathcal{N}_e(h_l)$ is the set of all edges that connect a node in $h_l$ and a node outside of $h_l$. $\deg_{h_l}(v_{l+1})$ specifies the number of nodes in $h_l$ that are connected to the node $v_{l+1}$.*

*Thus, the probability $\tilde{p}(x_k)$ of sampling a sequence $x_k = \{v_1, v_2, \cdots, v_k\} \in \mathcal{S}(g_k)$ in the subgraph expansion process is equal to*

$$\tilde{p}(x_k) = q(v_1) \prod_{l=1}^{k-1} \mathbb{P}(v_{l+1} \mid h_l) = \frac{f(\deg(v_1))}{F} \prod_{l=1}^{k-1} \frac{\left|E_{h_{l+1}}\right| - |E_{h_l}|}{\sum_{i=1}^{l} \deg(v_i) - 2|E_{h_l}|} \tag{27}$$

*Notice that*

$$p(g_k) = \sum_{x_k \in \mathcal{S}(g_k)} \tilde{p}(x_k) \tag{28}$$

*Since*

$$\mathcal{S}(g_k) = \{[v_1, \ldots, v_k] | \{v_1, \ldots, v_k\} = V_{g_k}, g_k | \{v_1, \ldots, v_l\} is\ connected\} \tag{29}$$

*,*

*then we have*

$$p(g_k) = \sum_{x_k \in \mathcal{S}(g_k)} \frac{f(\deg(x_k[1]))}{F} \prod_{l=1}^{k-1} \frac{\left| E_{h_{l+1}(x_k)} \right| - \left| E_{h_l(x_k)} \right|}{\sum_{i=1}^{l} \deg(x_k[i]) - 2 \left| E_{h_l(x_k)} \right|} \tag{30}$$

*where $x_k = \{v_1, v_2, \cdots, v_k\}$ be the original node ordering of $g_k$ via the subgraph expansion process. $x_k[i]$ be the $i^{th}$ node in $x_k$. $h_l(x_k) = \{v_1, v_2, \cdots, v_l, x_k, G\}$ be an induced subgraph of graph $G$ with the first l nodes $\{v_1, v_2, \cdots, v_l\}$ in $x_k$*

*Therefore, the proof is concluded.*

**Theorem 2.** *Let $\tilde{n}_{kr}(G) = \frac{1}{O} \sum_{o=1}^{O} \frac{\mathbb{I}(g_{ko} \sim \mathcal{G}_r)}{p(g_{ko})}$ be the estimation of graphlet counts, $d_1, \cdots, d_k$ be the k highest degrees of nodes in $G$, and denote $D = \prod_{l=2}^{k-1}(d_1 + \cdots + d_k)$. If q for sampling the starting node is the stationary distribution of the node random walk, then the upper bound of the variance $\text{Var}(\tilde{n}_{kr}(G))$ is*

$$\text{Var}(\tilde{n}_{kr}(G)) \leq \frac{1}{O} n_{kr}(G) \frac{2 \left| E_G \right|}{\left| \mathcal{S}(\mathcal{G}_r) \right|} D \tag{31}$$

*Proof. Consider sampling the starting node $v_1$ independently and from an arbitrary distribution q when we have access to all the nodes. Sampling nodes independently implies that the subgraph expansion process will result in independent k-subgraph samples. Thus, the variance of the graphlet count estimator can be decomposed into the variance of the individual k-subgraph samples. The variance of the estimator $\tilde{n}_{kr}(G)$ is then*

$$\text{Var}(\tilde{n}_{kr}(G)) = \frac{1}{O} \text{Var} \left( \frac{\mathbb{I}(g_{kO} \sim \mathcal{G}_r)}{p(g_{kO})} \right) = \frac{1}{O} \left( \sum_{g_k \in G_k} \frac{\mathbb{I}(g_k \sim \mathcal{G}_r)}{p(g_k)} - n_{kr}(G)^2 \right) \tag{32}$$

*It is observed that the variance $\text{Var}(\tilde{n}_{kr}(G))$ is small when the distribution of $p(g_k)$ is close to uniform distribution. A larger $p(g_k)$ results in a smaller variance of the estimator. Thus, the variation can be reduced by an appropriate choice of q for sampling the starting node, say a smaller normalizing factor F. In this case, the estimated graphlet count $\tilde{n}_{kr}(G)$ is close to the actual count $n_{kr}(G)$, which implies that the graphlet samples and all graphlets share similar distributions.*

*Let*

$$\phi_o = \frac{\mathbb{I}(g_{ko} \sim \mathcal{G}_r)}{p(g_{ko})} \tag{33}$$

*The variance can be rewritten as follows.*

$$\text{Var}(\tilde{n}_{kr}(G)) = \frac{1}{O} \text{Var}(\phi_o) \tag{34}$$

*Notice that $n_{kr}(G) = \mathbb{E}\phi_o$, and $\tilde{n}_{kr}(G) = \frac{1}{O} \sum_{o=1}^{O} \phi_o$ for the estimator.*

*We can bound the variancein Eq.(32) by the second moment, which is bounded by,*

$$\mathbb{E}\phi_o^2 \le \mathbb{E}\phi_o \max \phi_o = n_{kr}(G) \max \phi_o \tag{35}$$

*By seeking to control the the maximum of $\phi_o$, we have*

$$\max_{g_k} \frac{1}{p(g_k)} \le \max_{x_k} \frac{1}{|\mathcal{S}(g_k)|\tilde{p}(x_k)} \le \max \frac{\prod_{l=1}^{k-1}(d_1+\cdots+d_l)}{|\mathcal{S}(\mathcal{G}_r)|\,q(d_1)} \tag{36}$$

*and we obtain*

$$\max_x \frac{|\mathcal{N}_v(x)|}{\mathcal{S}(\mathcal{G}_r)|\tilde{p}(x)} \le \max \frac{\prod_{l=1}^{k-1}(d_1+\cdots+d_l)}{|\mathcal{S}(\mathcal{G}_r)|\,q(d_1)} \tag{37}$$

*Thus, we can construct a bound on $\mathrm{Var}\,(\phi_o)$ and $\mathrm{Var}\,(\tilde{n}_{kr}(G))$.*

*Therefore, the proof is concluded.*

**Theorem 3.** *Let $d$ be the dimension of the flattened $M_{b+1}$, $\otimes$ be an appropriate tensor product, $\mathcal{A}_{b+1} \in \mathbb{R}^{d\times d\times d}$ and $\mathcal{B}_{b+1} \in \mathbb{R}^{d\times d\times d\times d}$ are the tensors of $\mathcal{L}^s(M_{b+1})$ at iteration $b+1$ satisfying*

$$\mathcal{A}_{b+1} \otimes z_b^3 = \sum_{i,j,k=1}^{d} \frac{\partial^3 \mathcal{L}^s(M_{b+1})}{\partial M^i \partial M^j \partial M^k} z_b^i z_b^j z_b^k \tag{38}$$

*and*

$$\mathcal{B}_{b+1} \otimes z_b^4 = \sum_{i,j,k,l=1}^{d} \frac{\partial^4 \mathcal{L}^s(M_{b+1})}{\partial M^i \partial M^j \partial M^k \partial M^l} z_b^i z_b^j z_b^k z_b^l. \tag{39}$$

*Suppose that $\mathcal{L}^s(M_{b+1})$ is sufficiently smooth, if $||z_b||$ is small enough, then we have*

$$z_b^T \nabla^2 \mathcal{L}^s(M_{b+1})z_b - \alpha_{b+1}(\omega)z_b^T z_b = \left(\frac{1}{2} - \frac{\omega}{6}\right)\mathcal{A}_{b+1} \otimes z_b^3 - \left(\frac{1}{6} - \frac{\omega}{12}\right)\mathcal{B}_{b+1} \otimes z_b^4 + \mathcal{O}\left(\|z_b\|^5\right) \tag{40}$$

*Proof. By utilizing the Taylor expansion, we obtain*

$$\mathcal{L}^s(M_b) = \mathcal{L}^s(M_{b+1}) - (\nabla \mathcal{L}^s(M_{b+1}))^T z_b + \frac{1}{2}z_b^T \nabla^2 \mathcal{L}^s(M_{b+1})z_b - \\ \frac{1}{6}\mathcal{A}_{b+1} \otimes z_b^3 + \frac{1}{24}\mathcal{B}_{b+1} \otimes z_b^4 + \mathcal{O}\left(\|z_b\|^5\right) \tag{41}$$

*and*

$$(\nabla \mathcal{L}^s(M_b))^T z_b = (\nabla \mathcal{L}^s(M_{b+1}))^T z_b - z_b^T \nabla^2 \mathcal{L}^s(M_{b+1})z_b + \\ \frac{1}{2}\mathcal{A}_{b+1} \otimes z_b^3 - \frac{1}{6}\mathcal{B}_{b+1} \otimes z_b^4 + \mathcal{O}\left(\|z_b\|^5\right) \tag{42}$$

*In addition, we have*

$$\alpha_{b+1}(\omega) = \frac{z_b^T y_b + \omega \left[2\left(\mathcal{L}^s(M_b) - \mathcal{L}^s(M_{b+1})\right) + \left(\nabla \mathcal{L}^s(M_b) + \nabla \mathcal{L}^s(M_{b+1})\right)^T z_b\right]}{z_b^T z_b} \tag{43}$$

*By combining Eqs.(41), (42), and (43), we get*

$$
\begin{aligned}
&z_b^T \nabla^2 \mathcal{L}^s(M_{b+1}) z_b - \alpha_{b+1}(\omega) z_b^T z_b \\
=& z_b^T \nabla^2 \mathcal{L}^s(M_{b+1}) z_b - z_b^T y_b - \\
& \omega \left[ 2\left( \mathcal{L}^s(M_b) - \mathcal{L}^s(M_{b+1}) \right) + \left( \nabla \mathcal{L}^s(M_b) + \nabla \mathcal{L}^s(M_{b+1}) \right)^T z_b \right] \\
=& \left( \frac{1}{2} - \frac{\omega}{6} \right) \mathcal{A}_{b+1} \otimes z_b^3 - \left( \frac{1}{6} - \frac{\omega}{12} \right) \mathcal{B}_{b+1} \otimes z_b^4 + \mathcal{O}\left( \|z_b\|^5 \right)
\end{aligned}
\tag{44}
$$

*Therefore, the proof is concluded.*

Sensitivity analysis of weight $\omega$. Based on Eq.(43), if $\omega = 0$, we have

$$
\alpha_{b+1}(0) = \frac{z_b^T y_b}{z_b^T z_b}
\tag{45}
$$

Then, it derives the following equation based on Eq.(40).

$$
z_b^T \nabla^2 \mathcal{L}^s(M_{b+1}) z_b - \alpha_{b+1}(\omega) z_b^T z_b = \frac{1}{2} \mathcal{A}_{b+1} \otimes z_b^3 - \frac{1}{6} \mathcal{B}_{b+1} \otimes z_b^4 + \mathcal{O}\left( \|z_b\|^5 \right)
\tag{46}
$$

According to Eq.(46) and Theorem 3, it is reasonable to believe that if the weight parameter $\omega$ is chosen such that

$$
\left| \frac{1}{2} - \frac{\omega}{6} \right| < \frac{1}{2}
\tag{47}
$$

and

$$
\left| \frac{1}{6} - \frac{\omega}{12} \right| < \frac{1}{6}
\tag{48}
$$

i.e., $0 < \omega < 4$, then $\alpha_{b+1}(\omega) z_b^T z_b$ may capture the second order curvature $z_b^T \nabla^2 \mathcal{L}^s(M_{b+1}) z_b$ with a high precision.

Now, let us further compare several possible choices of $\omega$ and the corresponding formulas for $\alpha_{b+1}(\omega)$.

(1) If $\omega = 1$, then

$$
\alpha_{b+1}(1) = \frac{z_b^T y_b + 2\left( \mathcal{L}^s(M_b) - \mathcal{L}^s(M_{b+1}) \right) + \left( \nabla \mathcal{L}^s(M_b) + \nabla \mathcal{L}^s(M_{b+1}) \right)^T z_b}{z_b^T z_b}
\tag{49}
$$

The resulting matrix $\alpha_{b+1}(1)\mathbb{I}$ satisfies the weak quasi-Newton equation in Eq.(17). Based on Eq.(40), we have

$$
z_b^T \nabla^2 \mathcal{L}^s(M_{b+1}) z_b - \alpha_{b+1}(1) z_b^T z_b = \frac{1}{3} \mathcal{A}_{b+1} \otimes z_b^3 - \frac{1}{12} \mathcal{B}_{b+1} \otimes z_b^4 + \mathcal{O}\left( \|z_b\|^5 \right)
\tag{50}
$$

(2) If $\omega = 2$, then

$$
\alpha_{b+1}(2) = \frac{z_b^T y_b + 4\left( \mathcal{L}^s(M_b) - \mathcal{L}^s(M_{b+1}) \right) + 2\left( \nabla \mathcal{L}^s(M_b) + \nabla \mathcal{L}^s(M_{b+1}) \right)^T z_b}{z_b^T z_b}
\tag{51}
$$

The following equation is derived from Eq.(40).

$$z_b^T \nabla^2 \mathcal{L}^s(M_{b+1}) z_b - \alpha_{b+1}(2) z_b^T z_b = \frac{1}{6} \mathcal{A}_{b+1} \otimes z_b^3 + \mathcal{O}\left(\|z_b\|^5\right) \tag{52}$$

(3) If $\omega = 3$, then

$$\alpha_{b+1}(3) = \frac{z_b^T y_b + 6\left(\mathcal{L}^s(M_b) - \mathcal{L}^s(M_{b+1})\right) + 3\left(\nabla \mathcal{L}^s(M_b) + \nabla \mathcal{L}^s(M_{b+1})\right)^T z_b}{z_b^T z_b} \tag{53}$$

According to Eq.(40), we obtain

$$z_b^T \nabla^2 \mathcal{L}^s(M_{b+1}) z_b - \alpha_{b+1}(3) z_b^T z_b = \frac{1}{12} \mathcal{B}_{b+1} \otimes z_b^4 + \mathcal{O}\left(\|z_b\|^5\right) \tag{54}$$

**Theorem 4.** *Suppose $\|\nabla \mathcal{L}^s(M_b)\| \neq 0$, the solution $z_b$ of the separate trust region optimization* $\arg\min u_b(z) = \mathcal{L}^s(M_b) + (\nabla \mathcal{L}^s(M_b))^T z + \frac{1}{2} z^T \nabla^2 \mathcal{L}^s(M_b) z, \ s.t. \|z\| \leq \Delta^s$ *in Eq.(11) satisfies*

$$u_b(0) - u_b(z_b) \geq \frac{1}{2} \|\nabla \mathcal{L}^s(M_b)\| \min\left\{\Delta^s, \frac{\|\nabla \mathcal{L}^s(M_b)\|}{\alpha_b}\right\} \tag{55}$$

*Proof. We have the separate trust region optimization based on two weak quasi-Newton conditions as follows.*

$$z^* = \arg\min u_b(z) \approx \mathcal{L}^s(M_b) + (\nabla \mathcal{L}^s(M_b))^T z + \frac{1}{2} \alpha_b z^T z, \ s.t. \|z\| \leq \Delta^s \tag{56}$$

*Since $\|\nabla \mathcal{L}^s(M_b)\| \neq 0$, the solution of the separate trust region optimization based on two weak quasi-Newton conditions in Eq.(56) can be solved as follows.*

*(1) if $\|\nabla \mathcal{L}^s(M_b)\| \leq \alpha_b \Delta^s, z_b = -\frac{1}{\alpha_b} \nabla \mathcal{L}^s(M_b)$;*

*(2) if $\|\nabla \mathcal{L}^s(M_b)\| > \alpha_b \Delta^s$, the optimal solution $s_k$ will be on the boundary of the separate trust region, i.e., $z_b$ is the solution of the following problem.*

$$z^* = \arg\min u_b(z) \approx \mathcal{L}^s(M_b) + (\nabla \mathcal{L}^s(M_b))^T z + \frac{1}{2} \alpha_b z^T z, \ s.t. \|z\| = \Delta^s \tag{57}$$

*From Eq.(57), we have the solution $z_b = -\frac{\Delta^s}{\|\nabla \mathcal{L}^s(M_b)\|} \nabla \mathcal{L}^s(M_b)$.*

*Thus, the general solution of the separate trust region optimization based on two weak quasi-Newton conditions in Eq.(56) can be rewritten as follows.*

$$z_b = -\frac{1}{\tilde{\alpha}_b} \nabla \mathcal{L}^s(M_b), \quad \text{where } \tilde{\alpha}_b = \max\left\{\alpha_b, \frac{\|\nabla \mathcal{L}^s(M_b)\|}{\Delta^s}\right\} \tag{58}$$

*If $\|\nabla \mathcal{L}^s(M_b)\| \leq \alpha_b \Delta^s$, then $z_b = -\frac{1}{\alpha_b} \nabla \mathcal{L}^s(M_b)$. Thus, we obtain*

$$\begin{aligned}
u_b(0) - u_b(z_b) &= -(\nabla \mathcal{L}^s(M_b))^T \left(-\frac{1}{\alpha_b} \nabla \mathcal{L}^s(M_b)\right) \\
&\quad - \frac{1}{2} \left(-\frac{1}{\alpha_b} \nabla \mathcal{L}^s(M_b)\right)^T \alpha_b I \left(-\frac{1}{\alpha_b} \nabla \mathcal{L}^s(M_b)\right) \\
&= \frac{\|\nabla \mathcal{L}^s(M_b)\|^2}{\alpha_b} - \frac{1}{2} \frac{\|\nabla \mathcal{L}^s(M_b)\|^2}{\alpha_b} \\
&= \frac{1}{2} \frac{\|\nabla \mathcal{L}^s(M_b)\|^2}{\alpha_b}
\end{aligned} \tag{59}$$

*If $\|\nabla\mathcal{L}^s(M_b)\| > \alpha_b\Delta^s$, then $z_b = -\frac{\Delta^s}{\|\nabla\mathcal{L}^s(M_b)\|}\nabla\mathcal{L}^s(M_b)$. Hence, we have*

$$
\begin{aligned}
u_b(0) - u_b(z_b) =& -\left(\nabla\mathcal{L}^s(M_b)\right)^T\left(-\frac{\Delta^s}{\|\nabla\mathcal{L}^s(M_b)\|}\nabla\mathcal{L}^s(M_b)\right) \\
& -\frac{1}{2}\left(-\frac{\Delta^s}{\|\nabla\mathcal{L}^s(M_b)\|}\nabla\mathcal{L}^s(M_b)\right)^T\alpha_b I\left(-\frac{\Delta^s}{\|\nabla\mathcal{L}^s(M_b)\|}\nabla\mathcal{L}^s(M_b)\right) \\
=& \Delta^s\|\nabla\mathcal{L}^s(M_b)\| - \frac{1}{2}\alpha_b(\Delta^s)^2 \\
>& \Delta^s\|\nabla\mathcal{L}^s(M_b)\| - \frac{1}{2}\Delta^s\|\nabla\mathcal{L}^s(M_b)\| \\
=& \frac{1}{2}\Delta^s\|\nabla\mathcal{L}^s(M_b)\|
\end{aligned}
\tag{60}
$$

*By integrating Eqs.(59) and (60), we obtain Eq.(55).*

*Therefore, the proof is concluded.*

## A.3   ADDITIONAL EXPERIMENTS

**Final Performance and Convergence on SNS, PPI, and DBLP.** Table 3 and Figures-3-6 exhibit the quality of six centralized graph matching, ~~six federated graph learning,~~ and four federated domain adaption algorithms over SNS, PPI, and DBLP respectively, based on $Hits@1$, $Hits@5$, $Hits@10$, $Hits@50$, and $Loss$. Similar trends are observed for the comparison of federated graph matching effectiveness and convergence in these figures: our UFGM method achieves the close or much better than the centralized graph matching method, regarding $Hits@1$ ($>0.37$), $Hits@5$ ($>0.43$), $Hits@10$ ($>0.41$), and $Hits@50$ ($>0.45$) on three datasets respectively. Our UFGM method achieves better performance than all the competitors of ~~federated graph learning and~~ federated domain adaption in most experiments. In addition, our UFGM method can significantly speedup the convergence on two datasets in most experiments, compared with all federated learning algorithms. Especially, UFGM can converge around 1,000 training rounds and then always keep stable on SNS. This demonstrates that UFGM fully utilizes the proposed graphlet feature extraction techniques to generate the pseudo training data and employ the strength of supervised graph matching for accelerating the training convergence. The above experiment results demonstrate that UFGM is effective as well as efficient for addressing the federated graph matching problem. This advantage is very important for large-scale federated graph matching. For example, innovators were asked to develop privacy-preserving federated learning solutions that help tackle the challenge of international money laundering across large-scale local transaction network owned by multiple banks (NSF, 2022). Federated graph matching (FGM) can be utilized to infer cross-graph edges over multiple clients (e.g., identify the same potential criminals transferring money between multiple organizations) and derive a latent global graph (i.e., a global financial transaction network) (Suzumura et al., 2019; Wang et al., 2019a; Zhang et al., 2021a).

Table 3: Final performance on DBLP

| Type | Algorithm | $Hits@1$ | $Hits@5$ | $Hits@10$ | $Hits@50$ | $Loss$ |
|------|-----------|----------|----------|-----------|-----------|--------|
| Centralized Graph Matching | NextAlign | **0.572** | **0.609** | 0.632 | 0.690 | **0.222** |
| | NetTrans | 0.529 | 0.592 | 0.616 | 0.632 | 1.881 |
| | CPUGA | 0.136 | 0.199 | 0.276 | 0.296 | 2.232 |
| | ASAR-GM | 0.172 | 0.237 | 0.260 | 0.271 | 2.052 |
| | SIGMA | 0.276 | 0.360 | 0.378 | 0.421 | 1.992 |
| | SeedGNN | 0.530 | 0.582 | **0.637** | **0.702** | 4.185 |
| Federated Domain Adaption | DualAdapt | 0.000 | 0.001 | 0.001 | 0.001 | 4.023 |
| | EFDA | 0.000 | 0.000 | 0.000 | 0.000 | 2.452 |
| | WSDA | 0.000 | 0.001 | 0.001 | 0.001 | 3.332 |
| | FKA | 0.001 | 0.001 | 0.002 | 0.002 | 4.601 |
| | UFGM | 0.453 | 0.552 | 0.591 | 0.659 | 0.332 |

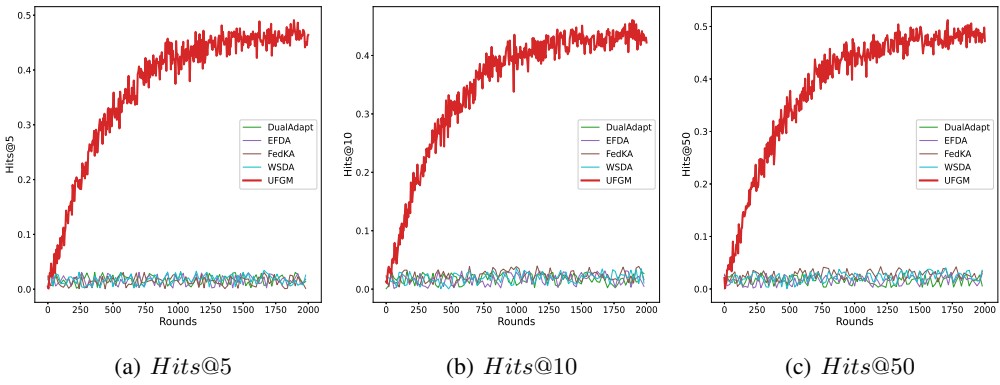

(a) $Hits@5$ (b) $Hits@10$ (c) $Hits@50$

Figure 3: Convergence on SNS

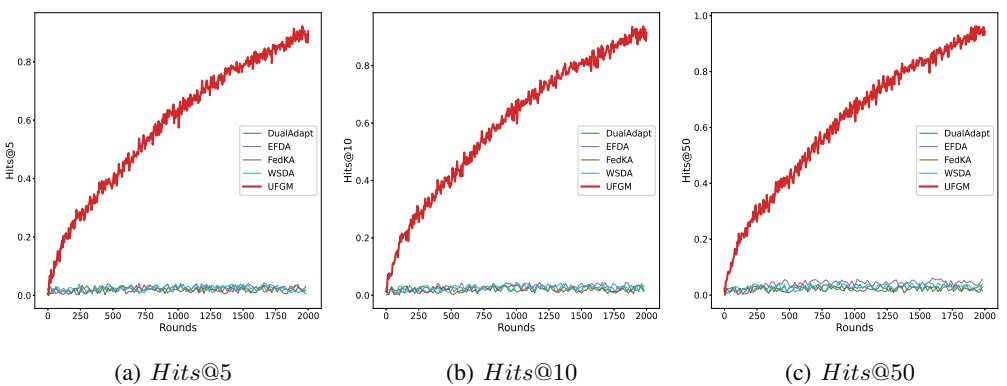

(a) $Hits@5$ (b) $Hits@10$ (c) $Hits@50$

Figure 4: Convergence on PPI

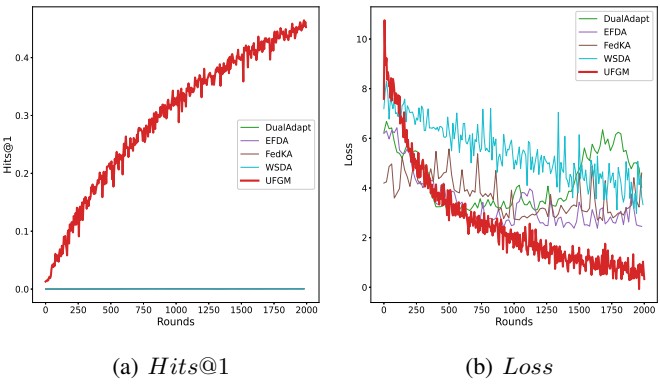

(a) $Hits@1$ (b) $Loss$

Figure 5: Convergence on DBLP

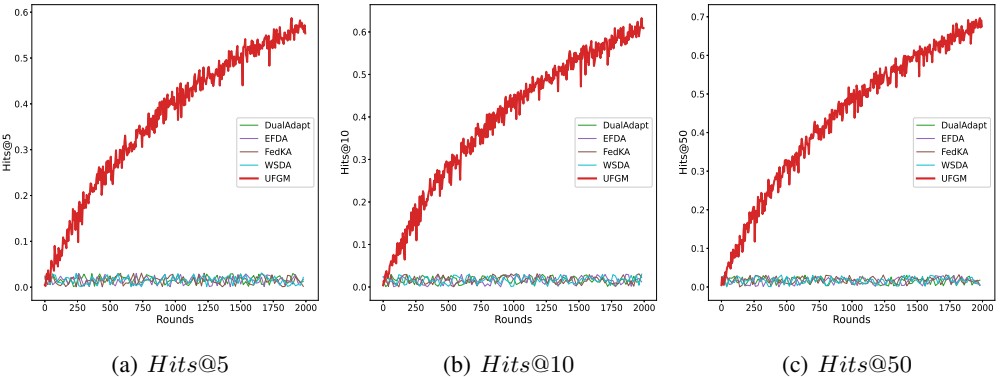

(a) $Hits@5$          (b) $Hits@10$          (c) $Hits@50$

Figure 6: Convergence on DBLP

Table 4: Final performance of centralized learning on SNS

| Algorithm | Dataset | $Hits@1$ | $Hits@5$ | $Hits@10$ | $Hits@50$ | $Loss$ |
|---|---|---|---|---|---|---|
| UFGM | SNS | 0.371 | 0.440 | 0.411 | 0.459 | 0.501 |
| | PPI | 0.771 | 0.880 | 0.902 | 0.930 | 0.659 |
| | DBLP | 0.453 | 0.552 | 0.591 | 0.659 | 0.332 |
| UFGM-C | SNS | 0.387 | 0.417 | 0.478 | 0.486 | 0.427 |
| | PPI | 0.786 | 0.911 | 0.922 | 0.932 | 0.495 |
| | DBLP | 0.471 | 0.563 | 0.635 | 0.718 | 0.182 |

**Final Performance of Centralized Learning.** We evaluate two versions of UFGM to show the strength of our UFGM method for federated graph matching. UFGM is the federated version with graph data encryption, graphlet feature extraction, model evaluation on the server, model optimization with the trust region on the clients, and Hessian approximation. UFGM-C is the centralized version with raw graph data uploaded to the server, graphlet feature extraction, model evaluation and model optimization with the standard stochastic gradient descent on the server. The experiment results in Table 4 exhibit that the performance of the centralized version, UFGM-C, is close to our federated version, UFGM over all three datasets. This further validates that our UFGM algorithm can achieve superior performance for the federated graph matching.

Table 5: Final performance of UFGM on large-scale datasets

| Dataset | $Hits@1$ | $Hits@5$ | $Hits@10$ | $Hits@50$ | $Loss$ |
|---|---|---|---|---|---|
| DBLP $100K$ | 0.536 | 0.659 | 0.671 | 0.735 | 1.115 |
| DBLP $200K$ | 0.405 | 0.496 | 0.559 | 0.619 | 1.720 |

**Final Performance on Large-scale Datasets.** In order to evaluate the scalability of our UFGM algorithm on large-scale datasets, we select and split the original DBLP dataset into 20 graphs by publication year, ranging from 2002-2022, such that each graph has around 100,000 and 200,000 authors as nodes and coauthor relationships as edges respectively. Thus, most authors occur in all 20 graphs but different graphs contain few emeritus and new authors. The experiment results in Table 5 demonstrate that our UFGM method scales well on two large-scale datasets.

Table 6: Final performance with new baselines on DBLP

| Type | Algorithm | $Hits@1$ | $Hits@5$ | $Hits@10$ | $Hits@50$ | $Loss$ |
|---|---|---|---|---|---|---|
| Unsupervised Centralized | GANN-GM | 0.034 | 0.058 | 0.082 | 0.126 | 4.125 |
| Graph Matching | REGAL | 0.349 | 0.425 | 0.472 | 0.551 | N/A |
| Unsupervised Federated | LADD | 0.002 | 0.003 | 0.004 | 0.011 | 4.120 |
| Domain Adaption | FMTDA | 0.008 | 0.011 | 0.016 | 0.029 | 1.597 |
| | UFGM | **0.453** | **0.552** | **0.591** | **0.659** | **0.332** |

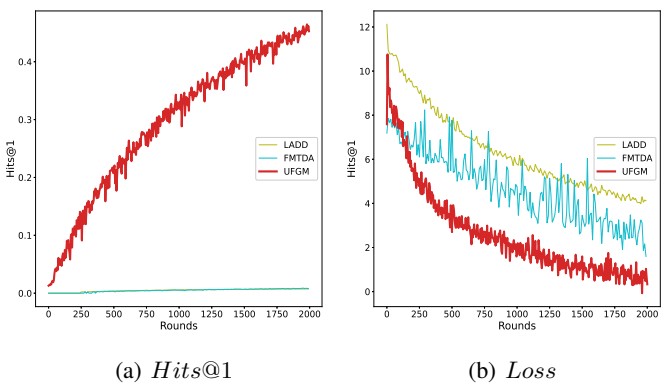

(a) $Hits@1$             (b) $Loss$

Figure 7: Convergence on DBLP

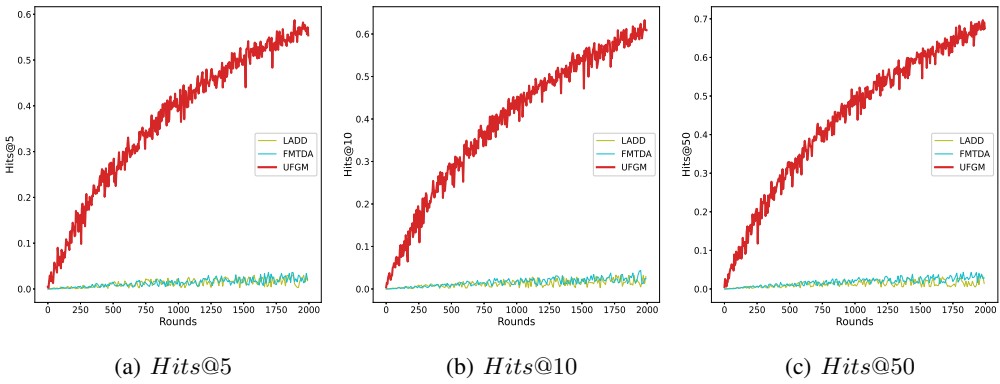

(a) $Hits@5$        (b) $Hits@10$        (c) $Hits@50$

Figure 8: Convergence on DBLP

**Final Performance and Convergence with New Baselines on DBLP.** Table 6 and Figures 7 and 8 exhibit the quality of our UFGM method with two unsupervised centralized graph matching approaches of **GANN-GM** (Wang et al., 2023) and **REGAL** (Heimann et al., 2018) and two unsupervised federated domain adaption algorithms of **LADD** (Shenaj et al., 2023) and **FMTDA** (Yao et al., 2022)~~, and two unsupervised federated graph learning methods of **FedWalk** (Pan & Zhu, 2022) and **Lumos** (Pan et al., 2023)~~. Similar trends are observed for the comparison among these unsupervised federated earning methods: our UFGM method outperforms these baselines in all experiments, in terms of both final performance and convergence. Notice that REGAL is a matrix factorization-based graph alignment method and thus there are no loss functions in it.

Table 7: Final performance of quasi-Newton approximation on three datasets

| Algorithm | Dataset | $Hits@1$ | $Hits@5$ | $Hits@10$ | $Hits@50$ | $Loss$ | Runing Time (m) |
|---|---|---|---|---|---|---|---|
| | SNS | 0.371 | 0.440 | 0.411 | 0.459 | 0.501 | 168 |
| UFGM | PPI | 0.771 | 0.880 | 0.902 | 0.930 | 0.659 | 153 |
| | DBLP | 0.453 | 0.552 | 0.591 | 0.659 | 0.332 | 732 |
| | SNS | 0.380 | 0.441 | 0.472 | 0.497 | 0.453 | 399 |
| UFGM-E | PPI | 0.786 | 0.907 | 0.937 | 0.947 | 0.633 | 367 |
| | DBLP | 0.487 | 0.606 | 0.657 | 0.687 | 0.228 | 1,556 |

**Final Performance of quasi-Newton Approximation.** We evaluate two versions of UFGM to show the strength of the quasi-Newton approximation for improving the efficiency while maintaining the quality federated graph matching. UFGM is the approximate version with the quasi-Newton approximation. UFGM-E is the exact version with the exact Hessian computation. The experiment results in Table 7 exhibit that the approximate version UFGM achieves slightly lower performance than the exact version UFGM-E but has much smaller running time. This demonstrates that the quasi-Newton approximation method is able to dramatically improve the efficiency while maintaining the utility constraints.

## A.4 PARAMETER SENSITIVITY

In this section, we conduct more experiments to validate the sensitivity of various parameters in our UFGM method for the federated graph matching task.

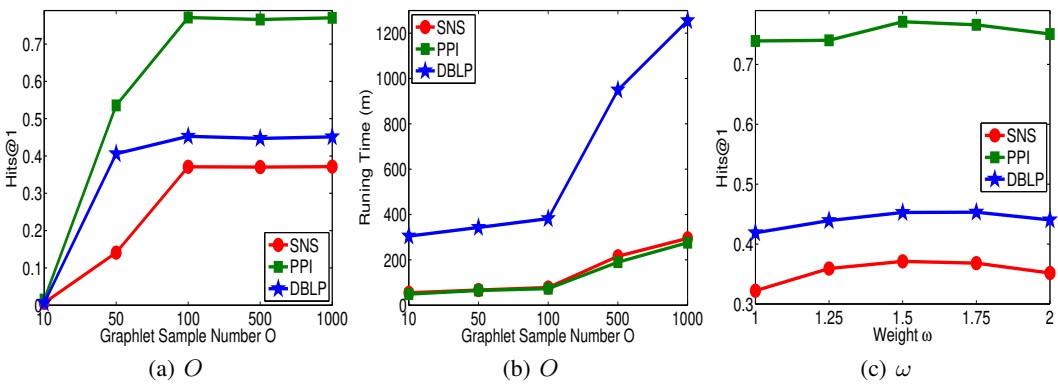

(a) $O$        (b) $O$        (c) $\omega$

Figure 9: Final $Hits@1$ with varying parameters on three datasets

**Impact of graphlet sample numbers.** Figure 9 (a) measures the performance effect of sampled graphlet numbers in the Monte Carlo Markov Chain sampling for graphlet enumeration and estimation by varying $O$ from 10 to 1,000. We have witnessed the performance curves by UFGM initially increase quickly and then become stable when $O$ continuously increases. Initially, a large $O$ can help utilize the strength of effective graphlet feature extraction for generating the pseudo training data for tackling the dilemma of unsupervised graph matching in federated setting and employing the strength of supervised graph matching. Later on, when $O$ continues to increase and goes beyond some thresholds, the performance curves become stable. A rational guess is that after the enough graphlet features have been already extracted at a certain threshold and considered in the FGM training, our UFGM model is able to generate a good graph matching result. When $O$ continuously increases, this does not affect the performance of graph matching any more. Figure 9 (b) reports the corresponding running time of our UFGM model by varying sampled graphlet number $O$ from 10 to 1,000. We make the observation on the quality and efficiency over three datasets: both the performance scores and the running time keep increasing when the sampled graphlet number is increasing. A rational guess is that a larger sampled graphlet number exchanges better performance with more sampling and processing time.

**Impact of weight $\omega$ between two types of weak quasi-Newton conditions.** Figures 9 (c) shows the influence of weight of two types of weak quasi-Newton conditions in our UFGM model by varying it from 1 to 2. It is observed that the performance initially raises when the $\omega$ increases. Intuitively, a large $\omega$ can help the algorithm well balance two types of weak quasi-Newton conditions and thus help improve the quality of separate trust region and graph matching. Later on, the performance curves decrease quickly when the $\omega$ continuously increases. A reasonable explanation is that a too large $\omega$ may ruin the first type of weak quasi-Newton condition and miss the optimal solution in the search process. Thus, it is important to determine the optimal $\omega$ for separate trust region.

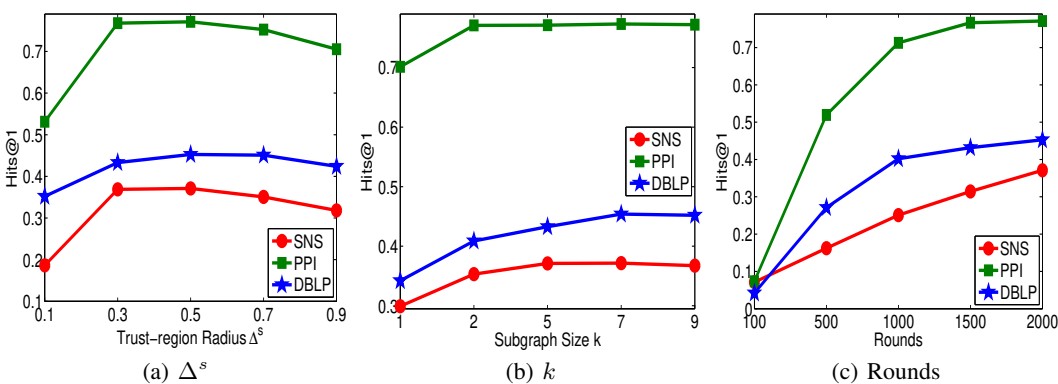

(a) $\Delta^s$          (b) $k$          (c) Rounds

Figure 10: Final $Hits@1$ with varying parameters on three datasets

**Influence of trust-region radius.** Figure 10 (a) demonstrates the influence of trust-region radius in the separate trust region in our UFGM model by varying it from 0.1 to 0.9. We have observed that the performance initially raises when the trust-region radius increases. Intuitively, a trust-region radius can help the algorithm quickly find the optimal solution and thus help improve the quality of federated graph matching. Later on, the performance curves decrease quickly when the trust-region radius continuously increases. A reasonable explanation is that a too large trust-region radius may miss the optimal solution with large step size in the search process. Thus, it is important to determine the optimal trust-region radius for the federated graph matching.

**Sensitivity of subgraph size.** Figure 10 (b) shows the influence of $k$-graphlets with $k$ nodes in the graphlet feature extraction in our UFGM model by varying it from 1 to 9. We make the observation: on the quality over three datasets: the performance curves keep increasing when the maximum subgraph size for the graphlet counting increases and then become stable when $k$ continuously increases. A rational guess is that a larger subgraph size initially makes UFGM capture more graphlet features and be more resilient to the unavailability of the training data. Later on, when $k$ continues to increase and goes beyond some thresholds, the performance curves become stable. A reasonable explanation is that after the enough graphlet features have been already extracted at a certain threshold and considered in the FGM training, our UFGM model is able to generate a good graph matching result. When $k$ continuously increases, this does not affect the performance of graph matching any more.

**Impact of training round.** Figure 10 (c) exhibits the sensitivity of training rounds of our UFGM model by varying them from 100 and 2,000. As we can see, the performance curves continuously increase with increasing training rounds. This is consistent with the fact that more training rounds makes the graph matching models be resilient to the federated setting. It is observed that our UFGM converges very fast on three datasets. From rounds 1,500 to 2,000, the $Hits@1$ scores oscillate within the range of 7.8% on three datasets.

Table 8: Final performance of quasi-Newton approximation on three datasets

| Pseudo Training Data | Dataset | $Hits@1$ | $Hits@5$ | $Hits@10$ | $Hits@50$ | $Loss$ |
|---|---|---|---|---|---|---|
| 20% | SNS | 0.077 | 0.116 | 0.176 | 0.292 | 0.557 |
|  | PPI | 0.372 | 0.440 | 0.497 | 0.627 | 0.669 |
|  | DBLP | 0.226 | 0.291 | 0.309 | 0.336 | 0.392 |
| 40% | SNS | 0.157 | 0.198 | 0.306 | 0.335 | 0.582 |
|  | PPI | 0.519 | 0.588 | 0.702 | 0.796 | 0.691 |
|  | DBLP | 0.312 | 0.378 | 0.397 | 0.442 | 0.449 |
| 60% | SNS | 0.302 | 0.332 | 0.347 | 0.407 | 0.512 |
|  | PPI | 0.628 | 0.776 | 0.825 | 0.917 | 0.686 |
|  | DBLP | 0.381 | 0.397 | 0.458 | 0.559 | 0.358 |
| 80% | SNS | 0.362 | 0.407 | 0.416 | 0.438 | 0.531 |
|  | PPI | 0.752 | 0.802 | 0.857 | 0.927 | 0.689 |
|  | DBLP | 0.406 | 0.497 | 0.533 | 0.610 | 0.349 |
| 100% | SNS | 0.371 | 0.440 | 0.411 | 0.459 | 0.501 |
|  | PPI | 0.771 | 0.880 | 0.902 | 0.930 | 0.659 |
|  | DBLP | 0.453 | 0.552 | 0.591 | 0.659 | 0.332 |

**Influence of pseudo training data.** Table 8 tests the influence of the pseudo training data for the performance of graph matching by varying the ratio of the pseudo training data from 20% to 100%. The ratio 100% corresponds to the number of the pseudo matched node pairs used in our current experiments. The numbers are 3,041 on SNS, 1,264 over PPI, and 2,817 on DBLP respectively. As we can see, the performance scores continuously increase with increasing pseudo training data. This is consistent with the fact that more training data makes the graph matching models achieve better performance.

## A.5 EXPERIMENTAL DETAILS

**Environment.** The experiments were conducted on a compute server running on Red Hat Enterprise Linux 7.2 with 2 CPUs of Intel Xeon E5-2650 v4 (at 2.66 GHz) and 8 GPUs of NVIDIA GeForce GTX 2080 Ti (with 11GB of GDDR6 on a 352-bit memory bus and memory bandwidth in the neighborhood of 620GB/s), 256GB of RAM, and 1TB of HDD. Overall, the experiments took about 2 days in a shared resource setting. We expect that a consumer-grade single-GPU machine (e.g., with a 2080 Ti GPU) could complete the full set of experiments in around 3-4 days, if its full resources were dedicated. The codes were implemented in Python 3.7.3 and PyTorch 1.0.14. We also employ Numpy 1.16.4 and Scipy 1.3.0 in the implementation. Since the datasets used are all public datasets and our methodologies and the hyperparameter settings are explicitly described in Section 3, 4, 5, and A.5, our codes and experiments can be easily reproduced on top of a GPU server. We promise to release our open-source codes on GitHub and maintain a project website with detailed documentation for long-term access by other researchers and end-users after the paper is accepted.

Table 9: Statistics of the datasets

| Dataset | #Clients/#Graphs | #Avg. Nodes | #Nodes | #Avg. Edges | #Edges |
|---|---|---|---|---|---|
| **SNS** | 3 | 14,331 | $14,262 \sim 14,573$ | 51,358 | $48,105 \sim 53,381$ |
| **PPI** | 50 | 1,767 | 1,767 | 32,320 | $31,179 \sim 32,358$ |
| **DBLP** | 20 | 10,038 | $9,984 \sim 10,168$ | 56,314 | $54,891 \sim 60,058$ |

**Datasets.** We study federated graph matching tasks on three representative graph matching benchmark datasets: social networks (SNS) [1], protein-protein interaction networks (PPI) [2], and DBLP coauthor graphs (DBLP) [3]. The above three graph datasets are all public datasets, which allow re-

---

[1] https://www.aminer.cn/cosnet

[2] http://snap.stanford.edu/ohmnet/

[3] http://dblp.uni-trier.de/xml/

searchers to use for non-commercial research and educational purposes. Among three datasets used in the experiment, social networks (SNS), protein-protein interaction networks (PPI), and DBLP coauthor graphs (DBLP) contain 3, 50, 20 different graphs respectively. These three datasets are widely used in training/evaluating the graph matching. The SNS dataset from (Zhang et al., 2015) has 3 different graphs of Flickr, Last.fm, and MySpace. The PPI dataset from (Zitnik & Leskovec, 2017) has 50 different graphs, each representing a tissue with proteins as nodes. As for the DBLP dataset, we select and split the original DBLP dataset into 20 graphs by publication year, ranging from 2002-2022. Thus, most authors occur in all 20 graphs but different graphs contain few emeritus and new authors.

**Training.** For each of the above three datasets, we use one client to maintain only one local graph in the federated setting. We randomly assign the graphs in the three datasets to 3, 50, 20 clients respectively in the experiments. We choose all of these graphs and clients to participate in the training of the models of federated graph matching. For the supervised learning methods, the training data ratio over the above three datasets is all fixed to 20%. We train the models on the training set and test them on the test set for three datasets. In addition, we run each experiment for 3 trials for obtaining more stable results.

**Baselines.** We compare three types of baselines that are most close to the task of federated graph matching: centralized graph matching, federated graph learning and federated domain adaption. (1) **Centralized graph matching baselines.** We compare the UFGM model with six state-of-the-art models. **NextAlign** is a semi-supervised network alignment method that achieves a good trade-off between alignment consistency and alignment disparity (Zhang et al., 2021c). **Net-Trans** is an end-to-end supervised graph matching model that learns a composition of nonlinear operations to transform one network to another in a hierarchical manner (Zhang et al., 2020). **CPUGA** is a robust supervised graph alignment model designed with non-sampling learning to distinguish noise from benign data in the given labeled data (Pei et al., 2022). **ASAR-GM** is a robust visual graph matching approach that enlarges the disparity among appearance-similar key-points in graph, orthogonal to de facto adversarial training (Ren et al., 2022). **SeedGNN** is a supervised approach that can learn from a training set how to match unseen graphs with only a few seeds (Yu et al., 2022). **SIGMA** is a semantIc-complete graph matching framework that completes mismatched semantics and reformulates the adaptation with graph matching (Li et al., 2022). (2) **Federated graph learning baselines.** We evaluate the UFGM model with six representative federated graph learning architectures. **FedGraphNN** is an open research federated learning system and a benchmark to facilitate GNN-based FL research (He et al., 2021a). **FKGE** is a decentralized scalable learning framework that learns knowledge graph embedding in an asynchronous and peer-to-peer manner while being privacy-preserving (Peng et al., 2021). **SpreadGNN** is a multi-task federated training framework capable of operating in the presence of partial labels and the absence of a central server for GNNs over molecular graphs (He et al., 2022). **SFL** is a structured federated learning framework to learn both the global and personalized models simultaneously using client-wise relation graphs and clients' private data (Chen et al., 2022b). **FederatedScope-GNN** is an easy-to-use FGL package that provides a unified view for modularizing and expressing FGL algorithms (Wang et al., 2022b). **FedStar** is an FGL framework that extracts and shares the common under-lying structure information for inter-graph federated learning tasks (Tan et al., 2022). (2) **Federated domain adaption baselines.** We compare the model performance with four recent federated domain adaption methods. **DualAdapt** aims to align the represen-tations learned among the different nodes with the data distribution of the target node (Peng et al., 2020). **EFDA** extends domain adaptation with the constraints of federated learning to train a model for the target domain and preserve the data privacy of all the source and target domains (Kang et al., 2022). **WSDA** leverages auxiliary information to reduce the risk of federated domain adaption on the target client during local training (Jiang & Koyejo, 2023). **FedKA** aligns features from different clients and those of the target task (Sun et al., 2022).

**Implementation.** For six state-of-the-art centralized graph matching models of NextAlign [4], Net-Trans [5], CPUGA [6], ASAR-GM [7], SeedGNN [8], and SIGMA [9], we used the open-source implementation and default parameter settings by the original authors for the experiments. All hyperparameters are standard values from reference codes or prior works. ~~For six representative federated graph learning architectures of FedGraphNN [10], FKGE [11], SpreadGNN [12], SFL [13], FederatedScope-GNN [14], and FedStar [15], we also use the default parameters in the authors' implementation.~~ For four recent federated domain adaption methods of DualAdapt [16], EFDA [17], WSDA [18], and FedKA [19], we utilized the same model architecture as the official implementation provided by the authors and used the same datasets to validate the performance of these federated graph matching models in all experiments. All models were trained for 2,000 rounds, with a batch size of 500, and a learning rate of 0.05. The above open-source codes from the GitHub are licensed under the MIT License, which only requires preservation of copyright and license notices and includes the permissions of commercial use, modification, distribution, and private use.

For our UFGM model, we performed hyperparameter selection by performing a parameter sweep on sampled graphlet numbers $O \in \{1, 5, 10, 15, 20\}$, weight of two types of weak quasi-Newton conditions $\omega \in \{1, 1.25, 1.5, 1.75, 2\}$, trust-region radius $\Delta^s \in \{0.1, 0.3, 0.5, 0.7, 0.9\}$, subgraph size for graphlet feature extraction $k \in \{1, 2, 5, 7, 9\}$, training round $\in \{100, 500, 1,000, 1,500, 2,000\}$, and learning rate $\in \{0.001, 0.005, 0.01, 0.05, 0.1, 0.5\}$. We select the best parameters over 50 epochs of training and evaluate the model at test time. In our current implementation, we first utilize an efficient matrix generation method (Randall, 1993) to produce a random nonsingular matrix $\mathbf{K}$ and then orthogonalize it to preserve the distances between the embedding vectors.

**Hyperparameter settings.**

Unless otherwise explicitly stated, we used the following default parameter settings in the experiments.

Table 10: Hyperparameter settings

| Parameter | Value |
|---|---|
| Training data ratio for supervised learning methods | 20% |
| Sampled graphlet numbers $O$ | 10 |
| Weight of two types of weak quasi-Newton conditions $\omega$ | 1.5 |
| Trust-region radius $\Delta^s$ | 0.5 |
| Subgraph size $k$ for graphlet feature extraction | 5 |
| Training round | 2,000 |
| Batch size for training the model | 500 |
| Learning rate | 0.05 |

[4] https://github.com/sizhang92/NextAlign-KDD21

[5] https://github.com/sizhang92/NetTrans-KDD20

[6] https://github.com/scpei/CPUGA

[7] https://github.com/Thinklab-SJTU/ThinkMatch

[8] https://openreview.net/forum?id=iYvbPx8GTta

[9] https://github.com/CityU-AIM-Group/SIGMA

[16] https://drive.google.com/file/d/1OekTpqB6qLfjlE2XUjQPm3F110KDMFc0/view?usp=sharing

[17] https://github.com/yuetan031/fedstar

[18] https://openreview.net/forum?id=_1gu0EX0mM3

[19] https://github.com/yuweisunn/federated-knowledge-alignment

