# OpenReview forum: "Unsupervised Federated Graph Matching with Graphlet Feature Extraction and Separate Trust Region"
_ICLR.cc/2024/Conference — Submitted to ICLR 2024_

### Official Review · Reviewer_3NNH · 2023-10-12

**Soundness:** 3 good
**Presentation:** 3 good
**Contribution:** 3 good
**Rating:** 6
**Confidence:** 3

**Summary:**

This paper proposes a method for unsupervised federated graph matching, i.e. matching node pairs that correspond to the same entity across different networks. The authors focus on the unsupervised setting in that it complies better with the privacy requirement. To address the problem of no ground truth matched pairs, the authors compute graphlet degree vectors of each node and use it to match cross network nodes. In addition, to ensure that clients do not know the matching information, the authors design a separate trust region algorithm, such that servers know the matching but does not know embeddings, and for clients vice versa. Some acceleration techniques are used to speedup the computation of Hessians (used in the trust region algorithm) and graphlet degree sampling.

Experiments are done over real-world network pairs, where the proposed method outperforms various FGL baselines. Parameter analysis experiments are also done.

**Strengths:**

1. The studied problem is new, novel and challenging. Graph node matching is an important task, and when it is put in federated learning, the authors made a good observation that supervised methods may compromise privacy. The observation is valid, and thus the studied problem, unsupervised federated graph matching, is of good practical value.

2. The design is also reasonable and practical. To ensure that the pseudo-matching information is not disclosed to clients, the authors design a separate trust region algorithm to split optimizations between client and server. Also, some acceleration techniques are proposed with theoretical results (although I did not thoroughly check their correctness).

3. The paper is well organized and states its design rationale in a clear manner. I have no problem following the paper.

4. Experiments are extensive considering the fact that the work does not have many baselines. The results are promising.

**Weaknesses:**

1. Despite the promising result, I have some questions regarding the fundamental assumption of this paper. The assumption is that, nodes across networks with similar graphlet degree vectors are likely to be the same nodes. However, as graph structures are different in different networks, it may well happen that a node's structure changes a lot, e.g. an author who changes his main research focus. Thus, I would like to know how accurate the assumption is. Maybe the authors can provide some numbers to show.

2. I wonder whether a split-learning technique can achieve the goal of the separate trust region algorithm. From my understanding, the separate trust region algorithm is designed so that the pseudo matchings are kept at servers and not exposed to clients. This may well be achieved with some kind of split learning, where clients maintain their graph feature encoders, and a matching unit is trained at the server with the pseudo labels. In this case, there would be even no requirement of the Hessian matrices. Can the authors justify the necessity of using the separate trust region algorithm and the Hessian?

3. I wonder what the effect is of the number of pseudo labels to the overall algorithm. Intuitively, this seems like a tradeoff, as when you include more pseudo labels, they tend to be less accurate and bring noise to the overall learning. I would like to see a result of such tradeoff.

4. I also wonder what are the effects of the monte-carlo markov chain sampling methods and the quasi-Newton methods on the overall method runtime/efficiency. The authors made no experiments to analyze the effectiveness either design.

5. It seems that the loss function is operated on the 'encrypted' embeddings $\hat{v}$ instead of $v$. At this time, a normal non-singular matrix $K$ may just twist some dimensions of the vectors, while shrink some other dimensions. Or in other worlds, a normal non-singular matrix $K$ does not preserve distance (which is exactly the loss function). Thus, how does the loss work on encrypted vector embeddings when they are applied a transformation that does not maintain distance is a little beyond me. Do you need orthogonal ones (those that maintain distance)?

**Questions:**

Q1. How accurate are the pseudo labels from graphlet degrees?

Q2. How necessary is the Hessian matrix? What if we use some sort of split learning?

Q3: What is the effect of the number of pseudo labels?

Q4: What are the effects of the MCMC and the quasi-Newton method on the overall efficiency?

Q5: Does a random non-singular method suffice in privacy-preservation and loss computation? Or do we need an orthogonal one?

---

> ### Author Response · Authors · 2023-11-22
> **Point-by-point response to the comments made by Reviewer 3NNH**
>
> We thank this reviewer for the great suggestion!
>
> **Weakness 1:** Despite the promising result, I have some questions regarding the fundamental assumption of this paper. The assumption is that, nodes across networks with similar graphlet degree vectors are likely to be the same nodes. However, as graph structures are different in different networks, it may well happen that a node's structure changes a lot, e.g. an author who changes his main research focus. Thus, I would like to know how accurate the assumption is. Maybe the authors can provide some numbers to show.
>
> **Answer:** Thanks for the constructive comments.
> A graphlet is a small graph of size up to $k$ nodes of a larger graph, such as triangle, wedge, or $k$-clique, which describes the local topology of a larger graph.
> A node's local topology can be measured by a graphlet feature vector, where each component denotes the frequency of one type of graphlets. Thus, a graphlet feature vector is one of node structure representation (Shervashidze et al., 2009; Kondor et al., 2009; Soufiani \& Airoldi, 2012; Jin et al., 2018; Tu et al., 2019).
> Based on structure, attribute, or embedding features, existing graph matching efforts often choose the node pairs with the smallest distances or largest similarities across the graphs are selected as the matching results (Man et al., 2016; Zhou et al., 2018a; Yasar \& Catalyurek, 2018; Li et al., 2019b;a; Chu et al., 2019; Fey et al., 2020).
> This submission just follows this mainstream strategy for structure-based graph matching to conduct graph matching based on the graphlet features.
> This work explores a general solution of federated graph matching for addressing the most common graph matching question: structure-based graph matching without considering other types of features.
>
> For the case of the graphs with quite different structures, a popular method is to combine other types of features to alleviate the structure inconsistency issue, say attribute-based graph matching. When other types of features are similar with only dissimilar structure, it is still possible that two copies of the same nodes in different graphs are still most similar to each other and can be identified as the matching results, as long as more other features are integrated or by reducing the weight of structure features.
>
>
>
>
> **Weakness 2 / Question 2:** I wonder whether a split-learning technique can achieve the goal of the separate trust region algorithm. From my understanding, the separate trust region algorithm is designed so that the pseudo matchings are kept at servers and not exposed to clients. This may well be achieved with some kind of split learning, where clients maintain their graph feature encoders, and a matching unit is trained at the server with the pseudo labels. In this case, there would be even no requirement of the Hessian matrices. Can the authors justify the necessity of using the separate trust region algorithm and the Hessian?
>
> How necessary is the Hessian matrix? What if we use some sort of split learning?
>
> **Answer:** Thanks for the kind suggestion.
> The time complexity of multiple graph matching typically increases exponentially with the number of graphs to be matched. If we choose to conduct both evaluation and optimization of the graph matching model at the server, then the computational capability of each client are idle. This will dramatically degrade the algorithm efficiency, especially large-scale graph matching is often time-consuming.
>
> Thus, we separate model optimization from model evaluation in the trust region algorithm: (1) the server aggregates the local model parameter on each client into a global model parameter, runs and evaluates the global parameter, and computes the individual loss and the first-order gradient for each client; (2) each client computes the second-order Hessian and optimizes the local model.
> The reason of using the second-order Hessian in the trust region algorithm is that the second-order Hessian can provide fast convergence in terms of number of training steps.
> The second-order Hessian computation and the local model optimization are most compute-intensive over large graphs. Thus, we move these two operations to the clients for making full use of the computational capability of each client to improve the efficiency, while maintaining the privacy constraints.

---

> > ### Author Response · Authors · 2023-11-22
> > **Point-by-point response to the comments made by Reviewer 3NNH**
> >
> > **Weaknesses 3 / Question 1 / Question 3:** I wonder what the effect is of the number of pseudo labels to the overall algorithm. Intuitively, this seems like a tradeoff, as when you include more pseudo labels, they tend to be less accurate and bring noise to the overall learning. I would like to see a result of such tradeoff.
> >
> > How accurate are the pseudo labels from graphlet degrees?
> >
> > What is the effect of the number of pseudo labels?
> >
> > **Answer**: Thanks for the comments.
> > Like other machine learning tasks, supervised graph matching methods usually achieve better performance than unsupervised graph matching approaches, since the former employs the strength of training data (i.e., pre-matched node pairs across graphs).
> > In this work, we propose to capture nodes' graphlet features to generate pseudo matched node pairs on different graphs across clients as the pseudo training data for leveraging the strength of supervised graph matching and improving the quality of unsupervised graph matching.
> > Thus, the pseudo training data has a positive effect on the graph matching results.
> >
> > In the revision, Table 8 tests the influence of the pseudo training data for the performance of graph matching by varying the ratio of the pseudo training data from 20\% to 100\%. The ratio 100\% corresponds to the number of the pseudo matched node pairs used in our current experiments. The numbers are 3,041 on SNS, 1,264 over PPI, and 2,817 on DBLP respectively. As we can see, the performance scores continuously increase with increasing pseudo training data. This is consistent with the fact that more training data makes the graph matching models achieve better performance.
> >
> > **Weaknesses 4 / Question 4:** I also wonder what are the effects of the monte-carlo markov chain sampling methods and the quasi-Newton methods on the overall method runtime/efficiency. The authors made no experiments to analyze the effectiveness either design.
> >
> > What are the effects of the MCMC and the quasi-Newton method on the overall efficiency?
> >
> > **Answer**: Thanks for your helpful comments.
> > In the revision, Figure 9 (b) reports the corresponding running time of our UFGM model by varying sampled graphlet number $O$ from 10 to 1,000. We make the observation on the quality and efficiency over three datasets: both the performance scores and the running time keep increasing when the sampled graphlet number is increasing. A rational guess is that a larger sampled graphlet number exchanges better performance with more sampling and processing time.
> >
> > In the revision, we evaluate two versions of UFGM to show the strength of the quasi-Newton approximation for improving the efficiency while maintaining the quality federated graph matching. UFGM is the approximate version with the quasi-Newton approximation. UFGM-E is the exact version with the exact Hessian computation. The experiment results in Table 7 exhibit that the approximate version UFGM achieves slightly lower performance than the exact version UFGM-E but has much smaller running time. This demonstrates that the quasiNewton approximation method is able to dramatically improve the efficiency while maintaining the utility constraints.
> >
> > **Weaknesses 5 / Question 5:** It seems that the loss function is operated on the 'encrypted' embeddings $\hat{v}$ instead of $v$. At this time, a normal non-singular matrix $K$ may just twist some dimensions of the vectors, while shrink some other dimensions. Or in other worlds, a normal non-singular matrix $K$ does not preserve distance (which is exactly the loss function). Thus, how does the loss work on encrypted vector embeddings when they are applied a transformation that does not maintain distance is a little beyond me. Do you need orthogonal ones (those that maintain distance)?
> >
> > Does a random non-singular method suffice in privacy-preservation and loss computation? Or do we need an orthogonal one?
> >
> > **Answer**: Thanks for your great comments. Sorry for any confusion caused. In our current implementation, we first produce a random nonsingular matrix $K$ and then orthogonalize it to preserve the distances between the embedding vectors. We have added this clarification to the revision.

---

> ### Comment · Reviewer_3NNH · 2023-11-22
> **Response acknowledged.**
>
> I have read the author response and find that they provide necessary information for me to answer my questions.
>
> I find that most of my questions are well answered. I am willing to recommend a weak accept at this point.
>
> I am still not fully convinced that split learning cannot do the job. As the authors say, the proposed method can utilize client side computation resources, but in the meantime, extra computation of Hessian is involved. Thus I still think that a well-designed split learning can achieve the goal without needing to compute Hessian. Yet this is just a minor question about the design choice, rather than the method itself.

---

> > ### Author Response · Authors · 2023-11-22
> > **Post-rebuttal Comments**
> >
> > We would like to thank you again for your valuable time and thoughtful and constructive comments.

---

### Official Review · Reviewer_SCKR · 2023-10-31

**Soundness:** 4 excellent
**Presentation:** 3 good
**Contribution:** 4 excellent
**Rating:** 8
**Confidence:** 5

**Summary:**

This paper is the first unsupervised federated graph matching solution for inferring matched node pairs on different graphs across clients while maintaining the privacy requirement of federated learning. The technical contributions of this work are very extensive/impressive. A key to the federated graph matching method is to secure data privacy. It proposes the data encryption and unsupervised learning to provide strong privacy protection. To enhance the matching quality, it develops the graphlet feature extraction and separate trust region for pseudo supervised learning for the problem of federated graph matching. Both theoretical and experimental analyses are shown to demonstrate the computation effectiveness of the proposed method. The paper is well-organized, contains enough information in a limited number of pages, and is easy to understand.

**Strengths:**

1.	Solving the graph matching problem in the environment of federated learning is of great importance in social networks and financial crime detection.
2.	The paper is the first to explore the potential of introducing federated learning to graph matching.
3.	Theoretical analysis about graphlet estimation and separate trust region within this work is novel and requires numerous technical developments.
4.	The experiments in the paper are extensive and convincing. The experimental results justify the effectiveness of the proposed method.
5.	The paper evaluates both centralized and federated variants of the method and most of federated results achieve comparable results to centralized baselines.

**Weaknesses:**

1.	The scale of the experiment is a bit small. What will the performance look like on large-scale dataset?
2.	Another concern is the benchmark comparison. Authors claim it is the first algorithm for federate graph matching, so it is better to emphasize how innovative compared with peer works.
3.	A minor issue is that the small font in the figure legend decreases the paper's readability.

**Questions:**

Scalability test and more discussions.

---

> ### Author Response · Authors · 2023-11-22
> **Point-by-point response to the comments made by Reviewer SCKR**
>
> We thank this reviewer for the encouraging comments. We are delighted to hear your positive comments on our contribution to the valuable research problem with significant real-world applications, such as financial crime detection, and the technical novelties in the FGM. The core requirement of FL is the privacy protection. The paper proposed model graphlet feature extraction and separate trust region optimization  and combine two techniques into a unified UFGM model. The proposed method capture nodes' graphlet features to generate pseudo matched node pairs on different graphs across clients as the pseudo training data for leveraging the strength of supervised graph matching  for improving the algorithm effectiveness. On the other hand, the combination of the unsupervised FGM and the encryption of local raw graph data is able to provide strong privacy protection for sensitive local data. In addition, the separate trust region for pseudo supervised FGM is helpful to enhance the efficiency while maintaining the privacy constraints.
>
> **Weakness 1:** The scale of the experiment is a bit small. What will the performance look like on large-scale dataset?
>
> **Answer**: Thanks for your great comments. In the appendix, in order to validate the scalability of our UFGM method for the problem of unsupervised federated graph matching on large-scale datasets, we selected and split the original DBLP dataset into 20 graphs by publication year, ranging from 2002-2022, such that each graph has around 100,000 and 200,000 authors as nodes and coauthor relationships as edges respectively. We have reported the $Hits@K$ scores in Table 5. The experiment results exhibit that our UFGM method scales well on different datasets . This demonstrates that our UFGM method is scalable for addressing the problem of unsupervised federated graph matching, while maintaining the privacy requirement of FL.
>
> **Weakness 2:** Another concern is the benchmark comparison. Authors claim it is the first algorithm for federate graph matching, so it is better to emphasize how innovative compared with peer works.
>
> **Answer**: Graph matching (i.e., network alignment) is one of the most important research topics in the graph domain, which aims to match the same entities (i.e., nodes) across two or more graphs. There is still a paucity of techniques of effective federated graph matching (FGM), which is much more difficult to study and has many real applications, such as user account linking in different social networks and financial crime detection. However, directly sharing and inferring matched node pairs on different graphs across clients and local graphs over multiple clients gives rise to a serious privacy leakage concern, which is the motivation of this work, compared with centralized graph matching methods.
>
> To our best knowledge, this work is the first to offer an unsupervised federated graph matching solution for inferring matched node pairs on different graphs across clients while maintaining the privacy requirement of FL, by leveraging the graphlet theory and trust region optimization. Our UFGM method exhibits three compelling advantages: (1) The combination of the unsupervised FGM and the encryption of local raw graph data is able to provide strong privacy protection for sensitive local data; (2) The graphlet feature extraction can leverage the strength of supervised graph matching with the pseudo training data for improving the matching quality; and (3) The separate trust region for pseudo supervised FGM is helpful to enhance the efficiency while maintaining the privacy constraints.

---

> > ### Comment · Reviewer_SCKR · 2023-11-22
> > **Official Comments by Reviewers**
> >
> > The reviewer would like to thank the authors for their detailed responses. Most of my concerns have been addressed, therefore I will keep my original score and vote for the acceptance.

---

> > > ### Author Response · Authors · 2023-11-22
> > > **Post-rebuttal Comments**
> > >
> > > We would like to thank you again for your valuable time and thoughtful and constructive comments.

---

### Official Review · Reviewer_TsnJ · 2023-11-01

**Soundness:** 4 excellent
**Presentation:** 3 good
**Contribution:** 4 excellent
**Rating:** 8
**Confidence:** 4

**Summary:**

This manuscript works on the first federated learning mechanism for graph matching with privacy maintainance that supports effective and efficient graph matching at the client and server level, by designing a fast approximate method for graphlet feature extraction for pseudo supervised learning, and by combining separate trust region algorithm with data encryption that satisfy with the privacy requirement of the federated learning. Specifically, the new method samples a small number of graphlets to capture graphlet features of each node as pseudo training data. At last, the method separates model optimization from model evaluation in the federated learning. In the empirical studies, results show it can achieve better performance than all federated learning baselines in all tests and obtain close or better performance than centralized graph matching method.

**Strengths:**

+Using graphlet feature extraction to generate pseudo training data is helpful to maintain the privacy constraint in the federated learning as well as leaverage the power of supervised learning for better quality.

+The incorporation of separate trust region into the federated learning algorithm for graph matching is interesting. The fact that convergence is also achieved in theory is well-done.

+The theoretical analysis of the approximation error and the convergence analysis seems novel and interesting. These theoretical resluts guranttee the effectiveness of federated graph matching in the context of unsupervised learning.

+The proposed task is well-motivated, the experiment result is promising, and the authors compare several different types of baselines to validate the superior performance of the proposed techniques.

**Weaknesses:**

-It seems the scope of the proposed method is specific as it seems to only be designed for federated graph matching. I wonder how hard it will be to generalize the proposed method to general federated learning.

-I can understand the limitations on the experimental side, however, it would be great to hear from authors regarding how the performance of the centralized variant of the proposed federated graph matching approach (i.e., no federated learning)? Does this approach have much better performance?

-Experiment figures are hard to follow due to small font size.

**Questions:**

See the weaknesses

---

> ### Author Response · Authors · 2023-11-22
> **Point-by-point response to the comments made by Reviewer TsnJ**
>
> We thank this reviewer for the helpful comments. We are delighted to hear your positive comments on our contribution to the unsupervised federated graph matching with privacy preservation and good performance. Following your precise assessment, we believe that our work makes a solid step to offer an unsupervised federated graph matching solution for inferring matched node pairs on different graphs across clients while maintaining the privacy requirement of federated learning. We also want to remark that the utilization of the pseudo supervised FGM is able to significantly improve the performance of FGM models.
>
> **Weaknesses 1:** It seems the scope of the proposed method is specific as it seems to only be designed for federated graph matching. I wonder how hard it will be to generalize the proposed method to general federated learning.
>
> **Answer**: Thanks for your helpful comments. Different from the FL for computer vision and other graph learning tasks, such as image classification and graph classification, where the data on a client are often independent of the data on another client and stochastic gradient descent (SGD) optimization can be used to evaluate and optimize the local FL models without the need of accessing the data on other clients. However, graph matching aims to match the same entities (i.e., nodes) across two or more graphs. Thus, the FGM needs to infer matched node pairs on different graphs across clients. Directly accessing and sharing local graphs over multiple clients gives rise to a serious privacy leakage concern.
> Therefore, the SGD optimization widely used in deep learning and FL fails to work on the clients in the FGM, since each client can access only its own local graph data and thus cannot update local loss based on the pseudo matched node pairs. We propose a separate trust region algorithm to separate local model optimization from model evaluation in the trust region algorithm for pseudo supervised FGM while maintaining the privacy constraints.
>
> **Weaknesses 2:** I can understand the limitations on the experimental side, however, it would be great to hear from authors regarding how the performance of the centralized variant of the proposed federated graph matching approach (i.e., no federated learning)? Does this approach have much better performance?
>
> **Answer**: Thanks for the kind suggestion. In the appendix, we evaluate two versions of UFGM to show the strength of our UFGM method for federated graph matching. UFGM is the federated version with graph data encryption, graphlet feature extraction, model evaluation on the server, model optimization with the trust region on the clients, and Hessian approximation. UFGM-C is the centralized version with raw graph data uploaded to the server, graphlet feature extraction, model evaluation and model optimization with the standard stochastic gradient descent on the server. The experiment results in Table 4 exhibit that the performance of the centralized version, UFGM-C, is close to our federated version, UFGM, over all three datasets. This further validates that our UFGM algorithm can achieve superior performance for the federated graph matching.
> In addition, the performance of our UFGM-C method is close to (NextAlign, NetTrans, and SeedGNN) or better than (CPUGA, ASAR-GM, and SIGMA) among six centralized graph matching methods. Notice that NextAlign, NetTrans, and SeedGNN are semi-supervised or supervised methods with 20\% training data.

---

### Official Review · Reviewer_D7Az · 2023-11-02

**Soundness:** 2 fair
**Presentation:** 2 fair
**Contribution:** 2 fair
**Rating:** 6
**Confidence:** 3

**Summary:**

The paper works on unsupervised federated graph matching. It proposes UFGM, where clients first train locally, then send encrypted node embeddings to the central server for aggregation. Theoretical analysis shows it can maintain good performance without expensive costs. Experiments show its performance.

**Strengths:**

1. The proposed method can solve such federated unsupervised graph matching problems.
2. Theoretical analysis is provided.
3. Experiments show its performance.

**Weaknesses:**

1. The challenges and applications of applying federated training on unsupervised graph matching are not clear.
2. The technical advancement of the method is unclear. Traditional graph-matching algorithms with encrypted aggregation on the server side can solve such a problem. Such encryption can be a huge computation and communication cost during training.
3. The algorithm comparison is unreasonable. There is an unreasonable number of comparison methods. All these federated methods are used for supervised training and should not be compared methods with unsupervised training.

**Questions:**

1. What is the key takeaway of the theoretical analysis? Can it guide the experiments?

---

> ### Author Response · Authors · 2023-11-22
> **Point-by-point response to the comments made by Reviewer D7Az**
>
> We thank this reviewer for the constructive comments.
>
> **Weakness 1:** The challenges and applications of applying federated training on unsupervised graph matching are not clear.
>
> **Answer:** Thanks for the thoughtful comment. Since FedAvg was proposed by Google in 2017 [1], federated learning (FL) has been a hot research topic in the field of machine learning.
> With the increasing regulation restrictions, user concerns, and commercial competition, the government agencies limit data collection and the users are not willing to share the data.
> The original intention of FL proposed by Google is to reduce privacy and security risks by limiting the attack surface to only the device, rather than the device and the cloud [1].
> Federated graph learning (FGL) has led to state-of-the-art innovations in various applications, such as node classification, graph classification, network embedding, and link prediction. There is still a paucity of techniques of effective federated graph matching (FGM), which is much more difficult to study.
>
> A good real-world application of FGM is that financial crime detection on transaction networks with millions to billions of bank customers and transactions [2-4]. Data exchange among clients and server about sensitive bank customer and transfer data should be limited for privacy risks.
> Especially, US and UK governments launched a privacy-enhancing technology (PET) challenge about federated learning for financial crime detection in July 2022 [5,6]. The dataset contains SWIFT transfer data between bank accounts and individual bank account transaction data. Frequent interbank transactions between the same or affiliated entities may be potential money laundering activities.
> Another real application is user account linking in different networks, since the user information contain many sensitive information. Directly collecting and uploading the raw user data to the server for centralized graph matching (CGM) gives rise to a serious privacy concern.
>
> [1] Communication-Efficient Learning of Deep Networks from Decentralized Data. AISTATS 2017.
>
> [2] Towards federated graph learning for collaborative financial crimes detection. CoRR, abs/1909.12946, 2019.
>
> [3] A semi-supervised graph attentive network for financial fraud detection. ICDM 2019.
>
> [4] Federated graph learning - A position paper. CoRR, abs/2105.11099, 2021.
>
> [5] https://research.ibm.com/blog/privacy-preserving-federated-learning-finance
>
> [6] https://www.drivendata.org/competitions/group/nist-federated-learning/
>
> The unsupervised graph matching fails to employ the strength of training data and thus often leads to sub-optimal solutions, compared with supervised graph matching. However, the latter using the pre-matched node pairs as the training data is improper for the FGM scenarios due to privacy risks of direct cross-graph information exchange.
>
> Thus, we propose to capture nodes' graphlet features to generate pseudo matched node pairs on different graphs across clients as the pseudo training data for leveraging the strength of supervised graph matching and improving the quality of unsupervised graph matching.
> However, graphlet enumeration one by one on large graphs is impossible due to expensive cost. We propose to leverage Monte Carlo Markov Chain (MCMC) technique for sampling a small number of graphlets.
>
> Stochastic gradient descent (SGD) optimization widely used in deep learning fails to work on the clients in the FGM, since each client can access only its own local graph data and thus cannot update local loss based on the pseudo matched node pairs.
> We propose a separate trust region algorithm for pseudo supervised FGM while maintaining the privacy constraints.
> We separate model optimization from model evaluation in the trust region algorithm: (1) the server aggregates the local model parameter on each client into a global model parameter, runs and evaluates the global parameter, and computes the individual loss and the first-order gradient for each client; (2) each client computes the second-order Hessian and optimizes the local model.

---

> > ### Author Response · Authors · 2023-11-22
> > **Point-by-point response to the comments made by Reviewer D7Az**
> >
> > **Weakness 2:** The technical advancement of the method is unclear. Traditional graph-matching algorithms with encrypted aggregation on the server side can solve such a problem. Such encryption can be a huge computation and communication cost during training.
> >
> > **Answer:** Thanks for the valuable suggestion. As discussed above, two main contributions of this paper are (1) The graphlet feature extraction can leverage the strength of supervised graph matching with the pseudo training data for improving the matching quality; and (2) The separate trust region for pseudo supervised FGM is helpful to enhance the efficiency while maintaining the privacy constraints.
> >
> > In the separate trust region optimization, the second-order Hessian computation and the local model optimization are most compute-intensive over large graphs. Thus, we let the server compute the individual loss and the first-order gradient and let the clients compute the second-order Hessians and optimize the local models. This design is helpful to make full use of the computational capability of each client to improve the efficiency, as large-scale graph matching is often time-consuming.
> >
> > The data encryption and communication is a one-time operation that happens before the training. Thus, its computation and communication cost is relatively trivial, compared with the cost of iterative model training and optimization.
> >
> >
> > **Weakness 3:** The algorithm comparison is unreasonable. There is an unreasonable number of comparison methods. All these federated methods are used for supervised training and should not be compared methods with unsupervised training.
> >
> > **Answer:** Thanks for the kind suggestion. In our experiments, three federated domain adaption baselines, including DualAdapt, EFDA, and FedKA, are unsupervised learning methods. As discussed above, supervised learning methods usually achieve much better performance than unsupervised learning approaches, since the former employs the strength of training data. In the experiments in the submission, our UFGM method achieves better performance than all supervised or unsupervised federated learning methods in most tests, and even obtains the close or better performance than the supervised centralized graph matching method.
> >
> > In the revision, Table 6 and Figures 7 and 8 exhibit the quality of our UFGM method with two unsupervised centralized graph matching approaches of GANN-GM (Wang et al., 2023) and REGAL (Heimann et al., 2018), two unsupervised federated domain adaption algorithms of LADD (Shenaj et al., 2023) and FMTDA (Yao et al., 2022), and two unsupervised federated graph learning methods of FedWalk (Pan \& Zhu, 2022) and Lumos (Pan et al., 2023). Similar trends are observed for the comparison among these unsupervised federated earning methods: our UFGM method outperforms these baselines in all experiments, in terms of both final performance and convergence. Notice that REGAL is a matrix factorization-based graph alignment method and thus there are no loss functions in it.
> >
> > **Question 1:** What is the key takeaway of the theoretical analysis? Can it guide the experiments?
> >
> > **Answer:** Thanks for the comment. Theorems 1 and 2 give an explicit solution of the probability of getting a $k$-subgraph via subgraph expansion and the variance of the estimation of graphlet counts. In addition, the theoretical analysis derives that the estimation variance can be reduced with a smaller normalizing factor. In this case, the estimated graphlet count is close to the actual count, which implies that the graphlet samples and all graphlets share similar distributions.
> > Theorems 3 and 4 derive the error introduced by the separate trust region due to the Hessian approximation and conduct the convergence analysis of the approximation method.

---

> ### Comment · Reviewer_D7Az · 2023-11-22
>
> I highly appreciate the point-by-point response!
> 1. The example “user account linking in different networks, since the user information contains many sensitive information” is reasonable for the motivation of federated graph matching.
> 2. The MCMC for graphlet feature extraction and separate trust regions can definitely solve such a problem.
>
> However, my concern about experiments is not resolved.
> 1. Could the authors elaborate more on why UFGM can be better than (semi)-supervised federated graph matching? Semi-supervised federated graph matching (eg. FL+NextAlign) should have the same accuracy as the centralized semi-supervised training by a few tricks (e.g. encrypts the node pairs). Could the authors elaborate more on how they run FedGraphNN, FKGE, SpreadGNN, SFL, FederatedScope-GNN, and FedStar?
> 2. Minor thing about the ablation study: Is graphlet feature extraction+separate trust regions (UFGM) better than graphlet feature extraction+fedavg?
>
> I increase my rating from 3 to 4 since the theory is definitely complete.

---

> > ### Author Response · Authors · 2023-11-23
> > **A kind reminder to update score in OpenReview system as promised during discussion phase, and further clarification**
> >
> > We would like to thank you again for your valuable time and thoughtful and constructive comments.
> >
> > To our best knowledge, this work is the first to offer an unsupervised federated graph matching solution. Thus, we choose three types of baselines that are most close to the task of federated graph matching: centralized graph matching, federated graph learning, and federated domain adaption. In our experiments, federated graph learning baselines, including FedGraphNN, FKGE, SpreadGNN, SFL, FederatedScope-GNN, and FedStar, are designed to address other graph learning  tasks, such as node classification, graph classification, and network embedding. Since graph matching has quite different objectives from other graph learning  tasks, our UFGM achieves better performance than these federated graph learning baselines.
> >
> > On the premise of maintaining privacy requirement in federated learning, it is difficult to tackle the problem of supervised federated graph matching, even with the encryption of training data, i.e., pre-matched node pairs. Before the encryption, among the clients and server, if a client owns the training data, then this client can use the training data to reconstruct the graphs on other clients, raising a serious privacy leakage concern. On the other hand, the server is not a data provider and thus cannot own the training data. In addition, the server cannot obtain the training data from the clients when the clients don't have the training data.
> >
> > In our experiments, federated graph learning baselines, are designed to address other graph learning  tasks, such as node classification, graph classification, and network embedding. We can obtain the node embeddings by all these baselines. Based on embedding features, we compute the similarity scores of pairwise node embeddings across graphs. The node pairs with the largest similarity scores are selected as the matching results.
> >
> > Stochastic gradient descent (SGD) optimization widely used in the FedAvg fails to work on the clients in the FGM, since each client can access only its own local graph data and thus cannot update local loss based on the pseudo matched node pairs. If we choose to conduct both evaluation and optimization of the graph matching model at the server, then the computational capability of each client are idle. This will dramatically degrade the algorithm efficiency, especially large-scale graph matching is often time-consuming.

---

> > > ### Comment · Reviewer_D7Az · 2023-11-23
> > >
> > > Thank you for the further clarification!
> > >
> > > According to "We can obtain the node embeddings by all these baselines.", how do these methods get such node embeddings? Is it just passing the local graph into a random GNN and getting such embedding? I would suggest author remove these baselines or clearly describe the implementation. The proposed method cannot directly compare with those methods.
> > >
> > > I would suggest author remove the statement that the unsupervised federated method can be better than the semi-supervised federated graph matching method (also there is no such method). It can be misleading for future researchers.
> > >
> > > I understand that the authors added those federated graph learning methods since there is no method for federated graph matching. I will lean to accept the paper if the authors remove those methods and those statements about supervised federated graph matching.

---

> ### Author Response · Authors · 2023-11-23
> **Post-rebuttal Comments**
>
> Thanks for the consideration of our comments!
>
> In previous versions of the submission, we use **federated graph learning** baselines to describe FedGraphNN, FKGE, SpreadGNN, SFL, FederatedScope-GNN, and FedStar, instead of term **federated graph matching**. Since these **federated graph learning** baselines are originally designed to address other graph learning tasks, such as node classification, graph classification, and network embedding., our UFGM achieves better performance than these **federated graph learning** baselines.
>
> In the current version of the submission, we have re-drawn Figures 1-8, updated Tables 1-3, and 6, and modified the corresponding descriptions to remove all experiment results of these **federated graph learning** baselines and all the statements about them. In order to satisfy the ICLR requirement that reflects new changes to the revision against the original submission, we currently use strikethroughs to represent the contents to be removed. We will physically remove the contents with the strikethrough representation in this version into the next revision.
>
> Thanks again for taking the time to review our submission!

---

### Author Response · Authors · 2023-11-22
**Common Comments to all Reviewers**

We would like to thank the four reviewers for the helpful and constructive comments. We have tried our best to clarify the concerns and comments by all four reviewers. We have presented the point-by-point response to the comments made by each of the four reviewers. We will include the discussions, analyses, explanations, and experiment results in this rebuttal into the next revision.

---

### Meta-Review · Program_Chairs · 2023-12-06

**Metareview:**

The authors study unsupervised federated graph matching and I agree with the reviewers that the problem is not well motivated. Citing references and a few-word description of application is not a strong enough motivation. I also find the first paragraph of the paper with excessive references unhelpful. Moreover, the authors fail to reference some of the main information-theoretic as well as algorithmic results pertaining to standard version of the problem (not the federated setting), such as work by Cullina et al., Ganassali et al., Ding and Du, etc….

PC/SAC Comment: After calibration and downweighting inflated and non-informative reviews, the decision is to reject at this time.

**Justification For Why Not Higher Score:**

I am not convinced the problem is really practically relevant nor I am blown by the theoretical findings.

**Justification For Why Not Lower Score:**

N/A

---

### Decision · Program_Chairs · 2024-01-16

Reject